# Single GPU Task Adaptation of Pathology Foundation Models for Whole Slide Image Analysis

**Neeraj Kumar**
Memorial Sloan Kettering Cancer Center
`kumarn6@mskcc.org`

**Chad Vanderbilt**
Memorial Sloan Kettering Cancer Center
`vanderbc@mskcc.org`

## Abstract

Pathology foundation models (PFMs) have emerged as powerful tools for analyzing whole slide images (WSIs). However, adapting these pretrained PFMs for specific clinical tasks presents considerable challenges, primarily due to the availability of only weak (WSI-level) labels for gigapixel images, necessitating multiple instance learning (MIL) paradigm for effective WSI analysis. This paper proposes a novel approach for single-GPU **T**ask **A**daptation of **PFM**s (TAPFM) that uses vision transformer (ViT) attention for MIL aggregation while optimizing both for feature representations and attention weights. The proposed approach maintains separate computational graphs for MIL aggregator and the PFM to create stable training dynamics that align with downstream task objectives during end-to-end adaptation. Evaluated on mutation prediction tasks for bladder cancer and lung adenocarcinoma across institutional and The Cancer Genome Atlas (TCGA) cohorts, TAPFM consistently outperforms conventional approaches, with H-Optimus-0 (TAPFM) outperforming the benchmarks. TAPFM effectively handles multi-label classification of actionable mutations as well. Thus, TAPFM makes adaptation of powerful pre-trained PFMs practical on standard hardware for various clinical applications.

## 1 Introduction

Hematoxylin and Eosin (H&E) staining is the most common slide preparation method in pathology, used for visualizing tissue architecture and cellular details for cancer diagnosis. Whole slide images (WSIs) serve as high-resolution digital representations of these tissue slides, commonly scanned at either $20\times$ or $40\times$ optical resolution that captures $0.50\mu^2$ or $0.25\mu^2$ of tissue per pixel, respectively. WSIs form the basis of computational pathology that employs machine learning (ML) and computer vision techniques for digital cancer assessment [1]. Due to memory constraints preventing direct processing of gigapixel WSIs and the availability of only slide-level labels for clinical tasks, WSI processing adopts a multiple instance learning (MIL) approach. In the MIL framework, each WSI is represented as a bag of smaller tiles (e.g., $224 \times 224 \times 3$); a neural network encodes each tile into a feature embedding, and the tile embeddings are then aggregated into a slide-level bag representation to compute predictions using only bag-level labels during training [2, 3], as illustrated in Figure 1.

Computational pathology has experienced a paradigm shift with the introduction of pathology foundation models (PFMs), which learn powerful representations from large collections of WSIs through self-supervised pre-training of vision transformers (ViTs) [4, 5, 6, 7, 8, 9]. For specific downstream applications such as gene mutation prediction, survival analysis, and treatment response estimation, existing methods typically use these PFMs as fixed feature extractors and train separate MIL aggregators to generate slide-level predictions [2, 10]. The fixed-feature approach fails to adapt PFM parameters to the specific downstream task, potentially limiting performance [11, 12, 13]. To address this limitation, we propose a novel **T**ask **A**daptation of **P**athology **F**oundation **M**odels (TAPFM) approach that: (1) leverages ViT's internal attention mechanism for MIL aggregation,

(2) maintains separate computational graphs for PFM and MIL parameter updates with a dual-loss mechanism on a single GPU, and (3) seamlessly integrates with popular PFMs to improve their performance on clinically relevant tasks.

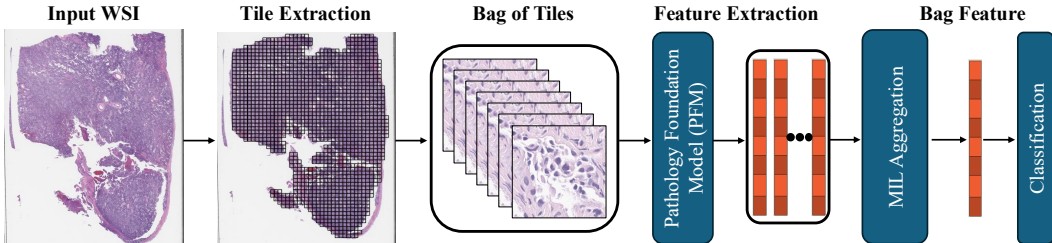

Figure 1: **WSI Processing Pipeline** – A representative H&E stained WSI scanned at $40\times$ optical resolution with $100,000 \times 125,000$ RGB pixels. For efficient processing, manageable sized tiles (e.g. $224 \times 224 \times 3$) shown in black boxes are usually extracted from the tissue region. Bag of tiles obtained from a WSI are then passed through a pathology foundation model to obtain feature representations (one feature vector per tile) which are then combined using multiple instance learning methods to compute a bag feature vector (one per WSI) that is used for downstream tasks such as binary classification (e.g. gene mutation prediction: 0 for absence and 1 for presence of mutation).

## 2 Related Work

### 2.1 Pathology Foundation Models (PFMs)

CTransPath [14] established an early benchmark by training a hybrid convolutional-transformer architecture on 32,220 WSIs across 25 anatomic sites. HIPT [15] and REMEDIS [16] explored different architectural approaches with ViT-S (DINO [17]) and ResNet-50 (SimCLR) respectively. Phikon [18] demonstrated the efficacy of ViT-L trained with iBOT on TCGA slides, while UNI [5] significantly expanded scale with its ViT-H architecture trained via DINOv2 [19] on 100,000 slides across 20 tissue types. Subsequent models pushed boundaries further with Virchow [6] exploring a ViT-huge model trained on 1.5 million WSIs. This trend toward increased scale continued with Prov-GigaPath [8] processing 1.3 billion tiles from 171,189 WSIs spanning 31 tissue types and Virchow2 [7] scaling to 1.7 billion tiles from 3.1 million slides across multiple magnifications. H-optimus-0 [9] leveraged ViT-giant architecture trained on hundreds of millions of tiles from over 500,000 WSIs. Several approaches have explored multimodality, including vision-language models (CONCH [20], PRISM [21], MUSK [22]) and vision-genomics integration (Orpheus [23]), expanding PFMs beyond visual representation learning. Despite this architectural diversity, vision-only models including UNI [5], GigaPath [8], and H-optimus-0 [9] have demonstrated superior performance on clinically relevant tasks such as cancer diagnosis, mutation prediction, and treatment response estimation [13]. The dominant architecture across these PFMs remains ViTs [4] trained through self-supervised learning, predominantly using DINOv2 [19], on diverse WSI data. *This paper specifically focuses on ViT based PFMs that utilize only pathology images as input.*

### 2.2 Multiple Instance Learning (MIL) in Computational Pathology

MIL methods for WSI analysis have evolved from attention-based mechanisms to spatial-aware architectures. Early MIL approaches used simple aggregation operations such as mean or max pooling to combine tile-level features [3]. A major advance was attention-based MIL (ABMIL) [2], which learns attention weights to prioritize diagnostically relevant tiles. CLAM [10] extended attention based MIL for multi-class classification. DSMIL [24] introduced a dual-stream approach coupling max-pooling with attention scoring. VarMIL [25] incorporated variance modeling to capture tissue heterogeneity while maintaining computational efficiency. Spatially aware MIL methods have also emerged to capture relationships between WSI tiles. TransMIL [26] leveraged transformer architectures with positional encoding, while graph-based approaches like PatchGCN [15] represent tiles as nodes in a graph structure based on physical adjacency. Graph transformer processing (GTP) [27] further refined this approach by combining graph structures with attention mechanisms. Despite architectural advances, benchmarking studies reveal that performance depends heavily on

the specific clinical task and the quality of input embeddings, with no single aggregation method consistently outperforming others across all applications [28].

## 2.3 Task Adaptation of PFMs

MIL methods usually rely on PFMs as fixed feature extractors, creating a disconnect between representation learning and task-specific adaptation for WSI analysis. Li et al. [29] proposed an Information Bottleneck (IB) based fine-tuning approach that addresses computational constraints through instance sparsification on smaller backbone models (ResNet-50). The multiple forward passes required by the IB approach make it computationally infeasible for modern large-scale PFMs on single GPU systems due to memory constraints. While recent approaches have attempted to avoid multiple forward passes through end-to-end fine-tuning of large-scale PFMs [30, 31], these methods typically require substantial computational resources spanning tens of GPUs. Most recently, Campanella et al. [32] demonstrated the clinical utility of PFM adaptation for *EGFR* mutation prediction in Lung cancer patients, but required distributed training across 24 NVIDIA H100 GPUs. Such computational requirements place PFM adaptation beyond the reach of most research groups and clinical institutions.

To the best of our knowledge, no existing approach has leveraged transformer's self-attention mechanism for MIL aggregation and enabled task adaptation of large-scale PFMs on a single GPU for downstream clinical applications, while addressing the optimization challenges that arise when jointly training foundation models and MIL aggregators.

## 3 Methodology

### 3.1 Problem Formulation

Let $X = \{x_1, x_2, ..., x_K\}$ represent an H&E stained WSI composed of $K$ non-overlapping tiles, where each tile $x_i \in \mathbb{R}^{H \times W \times C}$ corresponds to the tissue region extracted from the WSI. For MIL, a cancer patient's WSI has a bag-level label $y \in \{0, 1\}$ for binary classification tasks (e.g., presence or absence of a gene mutation in a patient), but no tile-level annotations. Our goal is to adapt a pretrained PFM $f_\theta$, parameterized by $\theta$, to extract tile-level features that are relevant for a downstream task. A ViT [4] based PFM maps each tile to a feature vector $z_i = f_\theta(x_i) \in \mathbb{R}^D$ [5, 8, 9] by first diving each input tile into a grid of $N$ non-overlapping patches (tokens) of size $P \times P$, with an additional learnable CLS token prepended to the sequence. These tokens are linearly projected and embedded with position information before being processed through multiple self-attention layers to compute tile feature representations.

### 3.2 Attention-Based Aggregation

The CLS token attends to all other tokens, to compute attention weights that indicate the importance of each token for the feature representation. We propose to leverage these attention weights for MIL aggregation. For a WSI with $K$ tiles, let $\mathbf{Z} = [z_1^T, z_2^T, \ldots, z_K^T]^T \in \mathbb{R}^{K \times D}$ denote the feature matrix, where each row $z_i \in \mathbb{R}^D$ is the feature vector (CLS token embedding) for tile $i$. Similarly, let $\mathbf{a} = [a_1, a_2, \ldots, a_K]^T \in \mathbb{R}^K$ denote the vector of attention weights derived from the ViT. For each tile $x_i$, the attention weight $a_i$ is computed as:

$$a_i = \frac{1}{H} \sum_{h=1}^{H} \frac{1}{N} \sum_{j=1}^{N} A_{cls,j}^h \tag{1}$$

where $A_{cls,j}^h$ is the attention weight from the CLS token to the $j^{th}$ token in the $h^{th}$ attention head while $H$ and $N$ are the numbers of attention heads and tokens, respectively. The proposed approach maintains a separate computation graph for the aggregator by detaching tile features and attention weights from the PFM's computation graph. The detached attention weights undergo min-max scaling to the range [0,1] followed by softmax normalization, ensuring they form a proper probability distribution for tile importance scoring to compute the bag representation $Z$ as:

$$Z = \mathbf{Z}^T \mathbf{a} = \sum_{i=1}^{K} a_i z_i \tag{2}$$

where both $\mathbf{Z}$ and $\mathbf{a}$ are detached from the PFM's computation graph. This ensures that the gradients from the classification loss only flow through the aggregator parameters while keeping the PFM's parameters fixed during this stage of optimization. The bag representation $Z$ is then passed through a linear classifier to predict the bag-level label:

$$\hat{y} = \sigma(WZ + b) \tag{3}$$

where $\sigma$ is the sigmoid activation while $W \in \mathbb{R}^{1 \times D}$ and $b \in \mathbb{R}$ are aggregator parameters $\theta_{agg}$ : $\{W, b\}$, that are learned during backpropagation with weighted cross-entropy loss:

$$\mathcal{L}_{agg}(y, \hat{y}) = -w_y[y \log(\hat{y}) + (1 - y) \log(1 - \hat{y})] \tag{4}$$

where $w_y$ is the weight assigned to class $y$ to handle class imbalance.

### 3.3 PFM Adaptation

Instead of employing conventional end-to-end backpropagation, we propose to detach gradients from the aggregator's computation graph to formulate a dedicated loss function for PFM adaptation.

**Feature Alignment Loss:** Let us define $\mathbf{G}_z = [g_{z_1}^T, g_{z_2}^T, \dots, g_{z_K}^T]^T \in \mathbb{R}^{K \times D}$ as the feature gradient matrix where $k^{th}$ row contains the gradient of the aggregator loss (equation 4) with respect to the corresponding tile's feature vector. During backpropagation through the aggregator, the gradients with respect to each feature vector are automatically computed (by chain rule) based on their contribution to the bag representation as $g_{z_i} = \frac{\partial \mathcal{L}_{agg}}{\partial z_i} = a_i \frac{\partial \mathcal{L}_{agg}}{\partial Z}$. We propose to detach the feature gradients from the aggregator's computation graph to compute the feature alignment loss as:

$$\mathcal{L}_{feature} = -\text{tr}(\mathbf{Z}\mathbf{G}_z^T) = \sum_{i=1}^{K} \langle z_i, g_{z_i} \rangle = \sum_{i=1}^{K} \sum_{d=1}^{D} z_{i,d} \times g_{z_i,d} \tag{5}$$

This loss guides feature vectors to move in the direction that reduces the classification loss and can be interpreted as a first-order approximation of the effect of feature changes on the aggregator loss.

**Attention Loss:** Let us define the gradient of the aggregator loss with respect to attention weights as $\mathbf{g}_a = [g_{a_1}, g_{a_2}, \dots, g_{a_K}]^T \in \mathbb{R}^K$. Similar to the feature gradients, the gradient of the aggregator loss with respect to each attention weight is automatically computed by the chain rule as $g_{a_i} = \frac{\partial \mathcal{L}_{agg}}{\partial a_i} = \langle z_i, \frac{\partial \mathcal{L}_{agg}}{\partial Z} \rangle$. We propose to compute the attention loss using the detached attention gradient as:

$$\mathcal{L}_{attention} = \mathbf{a}^T \mathbf{g}_a = \sum_{i=1}^{K} a_i \times g_{a_i} \tag{6}$$

This loss encourages attention weights to adjust based on the informativeness of each tile for the downstream task, increasing (or decreasing) weights for informative (or uninformative) tiles.

**Task Adaptation Loss (TAL):** For PFM updates, TAL combines the feature and the attention loss:

$$\mathcal{L}_{PFM} = \mathcal{L}_{feature} + \lambda \mathcal{L}_{attention} \tag{7}$$

where $\lambda$ is the hyperparameter that controls the relative importance of the attention loss. The PFM parameters, $\theta_{PFM}$, are then updated with the backpropagation using its own loss: $\mathcal{L}_{PFM}$. The training procedure for the proposed approach is presented in Algorithm 1 with illustraion shown in Figure 2 and its implementation is available at https://github.com/pfmadaptation/tapfm/.

### 3.4 Theoretical Analysis

The key innovation in our approach lies in the decoupling of the optimization process. In conventional end-to-end training with a unified computational graph, gradients flow through both the PFM and aggregator simultaneously:

$$\nabla_{\theta_{agg}, \theta_{PFM}} \mathcal{L} = \left( \frac{\partial \mathcal{L}_{agg}}{\partial \hat{y}} \cdot \frac{\partial \hat{y}}{\partial \theta_{agg}}, \frac{\partial \mathcal{L}_{agg}}{\partial \hat{y}} \cdot \frac{\partial \hat{y}}{\partial Z} \cdot \left( \frac{\partial Z}{\partial \mathbf{Z}} \cdot \frac{\partial \mathbf{Z}}{\partial \theta_{PFM}} + \frac{\partial Z}{\partial \mathbf{a}} \cdot \frac{\partial \mathbf{a}}{\partial \theta_{PFM}} \right) \right) \tag{8}$$

---

**Algorithm 1 T**ask **A**daptation of **P**athology **F**oundation **M**odels (TAPFM)

---

1: **Input:** Dataset $\mathcal{D}$ of WSIs and labels $\{(X_b, y_b)\}_{b=1}^{M}$, pretrained pathology foundation model $f_{\theta_{PFM}}$, MIL aggregator $f_{\theta_{agg}}$, learning rates ($\eta_{agg}$ and $\eta_{PFM}$)
2: **for** epoch = 1 to num_epochs **do**
3:    **for** each $(X_b, y_b)$ in $\mathcal{D}$ **do**
4:       Extract K tiles $\{x_1, x_2, ..., x_K\}$ from $X_b$ as tensor $\mathbf{X} \in \mathbb{R}^{H \times W \times C \times K}$
5:       // PFM forward pass
6:       $(\mathbf{Z}, \mathbf{A}) = f_\theta(\mathbf{X})$ {Extract last layer CLS feature and attention matrices}
7:       Compute $\mathbf{a} = [a_1, a_2, \ldots, a_K]^T \in \mathbb{R}^K$ where, $a_i = \frac{1}{H}\sum_{h=1}^{H}\frac{1}{N}\sum_{j=1}^{N} A_{cls,j}^h$ {Average CLS attention across heads and tokens}
8:       // Detach features and attention from PFM computation graph
9:       $\mathbf{Z}_{detached} = \text{detach}(\mathbf{Z})$ {Detach feature matrix}
10:      $\mathbf{a}_{detached} = \text{detach}(\mathbf{a})$ {Detach attention vector}
11:      // Aggregator parameter update
12:      $\mathbf{a}_{detached} = \text{softmax}(\text{minmax}(\mathbf{a}_{detached}))$ {Scale to [0,1] range and normalize}
13:      $Z = \mathbf{Z}^T \mathbf{a}$ {Compute Bag representation}
14:      Compute $\hat{y} = f_{\theta_{agg}}(z)$ and $\mathcal{L}_{agg}$ (equation 4)
15:      Backpropagate: $\theta_{agg} \leftarrow \theta_{agg} - \eta_{agg}\nabla_\theta \mathcal{L}_{agg}$
16:      // Detach gradients from aggregator computation graph
17:      $\mathbf{G}_z^{detached} = \text{detach}(\mathbf{G}_z)$ {Detach feature gradient matrix}
18:      $\mathbf{g}_a^{detached} = \text{detach}(\mathbf{g}_a)$ {Detach attention gradient vector}
19:      // PFM fine-tuning with detached gradients
20:      Compute fine-tuning loss $\mathcal{L}_{PFM}$ (Section 3.3)
21:      Backpropagate: $\theta_{PFM} \leftarrow \theta_{PFM} - \eta_{PFM}\nabla_{\theta_{PFM}}\mathcal{L}_{PFM}$
22:    **end for**
23: **end for**
24: **return** Fine-tuned PFM ($f_{\theta_{PFM}}$) and trained MIL aggregator ($f_{\theta_{agg}}$)

---

TAPFM instead implements the following two-stage optimization:

$$\nabla_{\theta_{agg}}\mathcal{L}_{agg} = \frac{\partial \mathcal{L}_{agg}}{\partial \hat{y}} \cdot \frac{\partial \hat{y}}{\partial \theta_{agg}} \quad \text{(Stage 1: Aggregator Update)} \tag{9}$$

$$\nabla_{\theta_{PFM}}\mathcal{L}_{PFM} = \text{detach}\left(\frac{\partial \mathcal{L}_{agg}}{\partial Z}\right) \cdot \left(\frac{\partial Z}{\partial \mathbf{a}} \cdot \frac{\partial \mathbf{a}}{\partial \theta_{PFM}} + \frac{\partial Z}{\partial \mathbf{Z}} \cdot \frac{\partial \mathbf{Z}}{\partial \theta_{PFM}}\right) \text{(Stage 2: PFM Update)} \tag{10}$$

This design resolves the circular dependency challenge in jointly optimizing PFM and MIL aggregator parameters by detaching gradients between optimization stages, that would otherwise create unstable training dynamics.

**Proposition 1** (Gradient Stabilization). *The TAPFM approach breaks the circular dependency at each iteration $t$ by enforcing:*

$$\frac{\partial \mathcal{L}_{agg}}{\partial \theta_{agg}}|_t \propto g(\theta_{PFM_{t-1}}, \theta_{agg_{t-1}}) \quad \text{and} \quad \frac{\partial \mathcal{L}_{PFM}}{\partial \theta_{PFM}}|_t \propto f(\theta_{PFM_{t-1}}, \theta_{agg_t}) \tag{11}$$

*resulting in more stable parameter trajectories than joint optimization.*

*Proof.* In joint optimization, the parameter updates create an implicit feedback loop:

$$\theta_{PFM_t} = \theta_{PFM_{t-1}} - \eta_{PFM}\nabla_{\theta_{PFM}}\mathcal{L}_{agg}(\theta_{PFM_{t-1}}, \theta_{agg_{t-1}}) \tag{12}$$

$$\theta_{agg_t} = \theta_{agg_{t-1}} - \eta_{agg}\nabla_{\theta_{agg}}\mathcal{L}_{agg}(\theta_{PFM_t}, \theta_{agg_{t-1}}) \tag{13}$$

Note that $\theta_{agg_t}$ depends on $\theta_{PFM_t}$, which itself depends on $\theta_{agg_{t-1}}$. This creates a circular dependency where each parameter set is chasing a moving target. However, the proposed approach breaks this loop by detaching the gradient computation graphs:

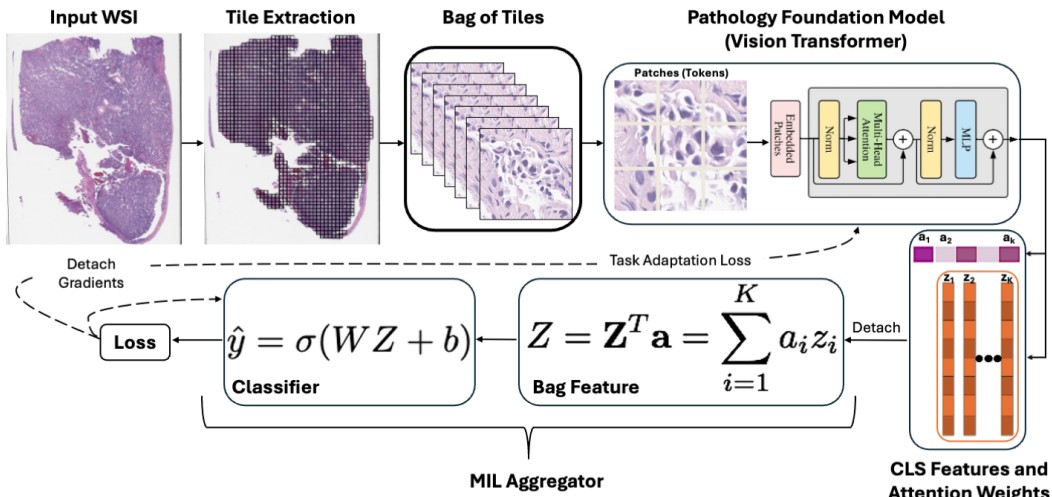

Figure 2: **TAPFM Algorithm Overview** – The proposed approach passes WSI tiles through a PFM to extract (last layer) CLS features $\mathbf{Z}$ and attention weights $\mathbf{a}$ (solid arrows). These are detached from the PFM's computation graph and used by the MIL aggregator to compute bag-level predictions and aggregator loss $\mathcal{L}_{agg}$ (solid arrows). During backpropagation (dashed arrows), gradients from $\mathcal{L}_{agg}$ are detached to formulate the task adaptation loss $\mathcal{L}_{PFM}$ (equation 7) for fine-tuning PFM parameters, while aggregator parameters are updated using $\mathcal{L}_{agg}$. This dual-optimization approach with separate computational graphs enables task adaptation on a single GPU.

$$\theta_{agg_t} = \theta_{agg_{t-1}} - \eta_{agg}\nabla_{\theta_{agg}}\mathcal{L}_{agg}(\text{detach}(\theta_{PFM_{t-1}}), \theta_{agg_{t-1}}) \tag{14}$$

$$\theta_{PFM_t} = \theta_{PFM_{t-1}} - \eta_{PFM}\nabla_{\theta_{PFM}}\mathcal{L}_{PFM}(\theta_{PFM_{t-1}}, \text{detach}(\theta_{agg_t})) \tag{15}$$

This detaching operation ensures that during aggregator optimization, $\theta_{PFM}$ is treated as a constant, and during PFM optimization, the updated $\theta_{agg}$ influences the loss but does not receive gradient updates. This effectively eliminates the circular dependency and stabilizes training. □

The stabilization effect of TAPFM's detached-gradient optimization is illustrated by the training-loss variance plots in Figure 5 in Appendix C. Figure 5 shows that when training with a unified computational graph, the loss oscillates without convergence, whereas TAPFM's detached-gradient optimization exhibits smooth exponential decay—confirming the theoretical stability analysis presented in this section. Additional properties of the proposed Algorithm 1 are elaborated in the appendix: permutation invariance (Appendix A.1), computational and space complexity analysis (Appendix A.2 and A.3), and prevention of catastrophic forgetting during task adaptation (Appendix A.4).

### 3.5 Multi-label Classification

The proposed Algorith 1 can be easily adapted for multi-label classification problems where each patient can be associated with multiple binary labels. Let $y = [y^1, y^2, ..., y^C]$ represent the ground truth vector where $y^j \in \{0, 1\}$ indicates the presence or absence of the $j$-th mutation. The aggregator loss for each class is computed using the weighted cross-entropy loss as in Equation 4. Note that the each output node in MIL aggregator is considered as class-specific binary predictor for multi-label classification. The total aggregator loss is then formulated as a weighted sum $\mathcal{L}_{agg} = \sum_{j=1}^{C} \alpha_j \mathcal{L}_{agg}^j$, where $\alpha_j$ is the weight assigned to class $j$, calculated as the inverse of the ratio of positive examples of that class to the total number of training instances.

Noticeably, multi-label classification differs fundamentally from both binary and multi-class approaches. While multi-class classification assumes mutual exclusivity among classes (one-hot encoding), multi-label classification allows multiple concurrent positive labels, better reflecting clinical reality where tumors can harbor multiple actionable mutations simultaneously [33]. This

is particularly relevant in scenarios such as collision tumors [34] and resistance-driven secondary mutations [35], where combinations of mutations have distinct prognostic implications [36].

While the methodology presented in this paper primarily focuses on classification tasks, TAPFM's core approach generalizes to other clinically relevant objectives as well. By replacing the classification head (Equation 4) with task-specific layers (e.g., Multi-task logistic regression for survival prediction [37]) and corresponding loss functions, TAPFM can adapt PFMs for survival analysis tasks while maintaining single-GPU training efficiency. Additional survival prediction experiments illustrating this generalization are provided in Appendix D.2.

# 4 Experiments

The proposed TAPM approach is evaluated on clinically relevant mutation prediction tasks using institutional and public cohorts of bladder cancer (BLCA) and lung adenocarcinoma (LUAD) patients.

**Datasets**: The institutional dataset consists of H&E WSIs of $2,030$ BLCA and $8,820$ LUAD patients collected during routine clinical care at Memorial Sloan Kettering Cancer Center (MSKCC), including both $20\times$ and $40\times$ ($\sim 30\%$ of all WSIs in each cohort) magnifications to reflect real-world data acquisition variability. WSIs from The Cancer Genome Atlas (TCGA) cohorts – TCGA-BLCA (260 patients) and TCGA-LUAD (438 patients), all scanned at $40\times$ resolution, are exclusively used for external validation to assess generalizability of the proposed approach. Only one WSI per patient is used in all training and validation cohorts.

For the binary classification task, TAPM is evaluated on two clinically relevant mutation prediction tasks: *FGFR3* in BLCA and *EGFR* in LUAD. For multi-label classification, TAPM's ability to simultaneously predict four actionable mutations in LUAD patients is assessed: *EGFR*, *KRAS*, *MET*, and *ALK*. The prevalence rate of these mutations across institutional (TCGA) cohorts are: 16% (14%) for *FGFR3* in BLCA, and in LUAD: 26% (14%) for *EGFR*, 27% (35%) for *KRAS*, 4% (2%) for *MET*, and 3% (1%) for *ALK*.

**Benchmarks**: We evaluate state-of-the-art PFMs such as **UNI** [5], **GigaPath** [8], and **H-Optimus-0** [9]. Motivated by prior findings that lightweight MIL models can attain clinical performance comparable to computationally expensive aggregators [28], this work employs memory-efficient MIL methods on single-GPU systems: **ABMIL** [2], **DSMIL** [24], **CLAM** [10], and **VarMIL** [25].

**Implementation Details**: All reported experiments in this paper are conducted using PyTorch 2.5.1 on a single NVIDIA H100 (80GB memory). Each WSI is processed at native resolution ($20\times$ or $40\times$) to extract non-overlapping tiles in a sliding window manner after filtering out the non-tissue regions with Otsu thresholding [38]. For $20\times$ WSIs, tiles of size $224 \times 224 \times 3$ pixels are extracted directly, while for $40\times$ images, tiles of size $448 \times 448 \times 3$ pixels are extracted and resized to $224 \times 224 \times 3$ pixels to maintain consistent spatial context. As detailed in the space complexity analysis (Appendix A.3), the memory requirements of the proposed TAPFM method scale quadratically with the number of tokens (patches) processed by ViTs. Thus, the $224 \times 224$ tile size is selected as the optimal dimension that prevents out-of-memory errors while maximizing the contextual information captured per tile. During training, 300, 100, and 75 tiles per WSI per epoch are randomly sample without replacement for UNI, Gigapath, and H-Optimus-0 respectively – the maximum number of tiles processable for each PFM on a single H100 GPU while maintaining end-to-end fine-tuning capability. *At inference, all tiles obtained from a given WSI are used for downstream mutation prediction tasks.*

For the proposed TAPFM method $\lambda = 1.0$ is used for all experiments. The institutional cohort is evaluated under 5-fold cross-validation at the patient level. In each fold, patients are stratified by label and scan resolution and split into 80% training, 10% validation, and 10% testing, with no patient overlap across splits. Area Under the receiver operating characteristics Curve (AUC) is the primary metric reported (binary: AUC; multi-label: macro-average AUC). Model selection is performed on the fold-specific validation set. For external validation, each of the 5 fold-specific models trained on the institutional data is evaluated on the corresponding TCGA cohort.

For training, AdamW [39] with weight decay of 1e-4 is used as opitimzer, applying differential learning rates of 1e-6 and 1e-5 for PFM and aggregator parameters, respectively. Training data augmentation included random horizontal flips, random rotations (of 90°, 180°, or 270°), and Gaussian blur. A cosine annealing scheduler with warm restarts ($T_0 = 10, T_{mult} = 2$) is also used for better convergence [40]. Each batch contained 1 WSI, and TAPFM is trained for 20 epochs while

all other benchmarks are trained for 50 epochs, with the institutional validation set used to select the best-performing model for all evaluations on the insititutional and TCGA testing sets.

## 4.1 Results

Table 1 shows the performance comparison of three sets of models on the testing data: (1) fixed-PFM with trained MIL aggregators, (2) fine-tuned PFM and MIL aggregators (equivalent to setting $\lambda = 0$ in equation 7 with external MIL methods), and (3) proposed TAPFM. It is evident that the proposed TAPFM approach outperforms the other benchmarks across both binary mutation prediction tasks. H-Optimus-0 (TAPFM) consistently achieves the best performance across both institutional and external TCGA testing cohorts, followed by Gigapath (TAPFM), indicates generalizability of the proposed approach. Table 2 extends TAPFM evaluation to the more challenging task of simultaneously predicting four actionable mutations in LUAD. H-Optimus-0 (TAPFM) consistently outperforms GigaPath (TAPFM) across all mutations, even for the rate *MET* and *ALK* mutations. Area Under Precision-Recall Curve (AUPRC) values are reported in Appendix D.1.

## 4.2 Runtime Performance

Training times for BLCA are 12 hours for UNI, 21 hours for Gigapath, and 24 hours for H-Optimus-0. For LUAD cases, training required 2 days 4 hours for UNI, 4 days 2 hours for Gigapath, and 4 days 6 hours for Hoptimus. Inference times per WSI are 4.85 minutes for UNI, 6.38 minutes for Gigapath, and 7.15 minutes for H-Optimus-0. These results confirm that TAPFM enables efficient PFM task adaptation on standard hardware making it suitable for clinical implementation.

## 4.3 Convergence

For binary *FGFR3* classification in BLCA, Figure 3a shows UNI reaching maximum validation performance by epoch 6 (AUC 0.8542), Gigapath by epoch 5 (AUC 0.8764), and H-Optimus-0 by epoch 7 (AUC 0.8960). Additionally, experiments on LUAD datasets (not shown) demonstrated that all PFMs converge within 4 epochs for binary classification tasks. For multi-label LUAD classification, convergence occurred at epoch 8 for UNI, epoch 10 for Gigapath and epoch 11 for H-Optimus-0. Loss-variance comparisons between joint optimization and TAPFM's detached-gradient training are shown in Appendix 5, illustrating markedly smoother convergence and confirming the theoretical stability analysis presented in Section 3.4.

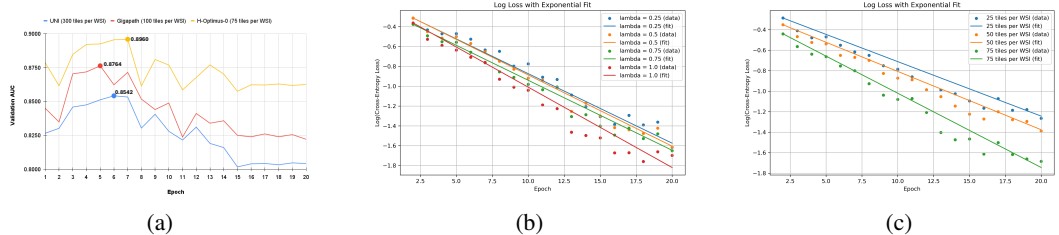

(a)  (b)  (c)

Figure 3: For bianry *FGFR3* prediction task in BLCA – (a) validation AUC trajectories of various TAPFM models, and the variation of log-transformed training loss of the H-Optimus-0 (TAPFM) model with (b) attention loss weighting ($\lambda$) and (c) tile sampling density.

## 4.4 Ablation Studies

Systematic ablation studies investigate the influence of key hyperparameters – attention loss weighting ($\lambda$ in equation 7) and number of tiles sampled per WSI per epoch – on the performance of H-Optimus-0 (TAPFM) model for the binary *FGFR3* prediction task in BLCA patients.

**Lambda**: To evaluate the impact of the attention loss term in TAL (Equation 7), log-linear convergence fits of the form $\log \mathcal{L}_k = a + b\,k$ were computed over epochs $k = 2$ to $20$. Log-transformed loss fits are shown in Figure 3b, with raw curves in Appendix 4a. All values of $\lambda$ yielded consistent exponential decay, with convergence rates of $b = -0.0701$ ($R^2 = 0.9737$), $-0.0718$ ($R^2 = 0.9808$), $-0.0706$ ($R^2 = 0.9765$), and $-0.0810$ ($R^2 = 0.9693$) for $\lambda \in \{0.25, 0.5, 0.75, 1.0\}$, respectively.

Table 1: Performance comparison of different PFM and MIL aggregation methods for binary classification tasks in terms of AUC (mean $\pm$ 95% CI from 5-fold cross-validation). $N$ indicates the number of patients in each cohort. Best and second best models are in bold and underlined text, respectively.

| Model | BLCA *FGFR3* | | LUAD *EGFR* | |
|---|---|---|---|---|
| | **Institutional Cohort** ($N = 194$) | **TCGA** ($N = 260$) | **Institutional Cohort** ($N = 876$) | **TCGA** ($N = 438$) |
| *Fixed-PFM with Trained MIL Aggregators* | | | | |
| UNI + DSMIL | $0.7876 \pm 0.024$ | $0.7928 \pm 0.022$ | $0.7352 \pm 0.021$ | $0.7624 \pm 0.025$ |
| UNI + CLAM | $0.7893 \pm 0.023$ | $0.7912 \pm 0.024$ | $0.7389 \pm 0.022$ | $0.7645 \pm 0.024$ |
| UNI + VarMIL | $0.7912 \pm 0.022$ | $0.7948 \pm 0.021$ | $0.7414 \pm 0.020$ | $0.7663 \pm 0.023$ |
| UNI + ABMIL | $0.7904 \pm 0.023$ | $0.7962 \pm 0.023$ | $0.7396 \pm 0.021$ | $0.7718 \pm 0.022$ |
| GigaPath + DSMIL | $0.8232 \pm 0.021$ | $0.8645 \pm 0.019$ | $0.7657 \pm 0.020$ | $0.8153 \pm 0.022$ |
| GigaPath + CLAM | $0.8246 \pm 0.020$ | $0.8673 \pm 0.018$ | $0.7672 \pm 0.019$ | $0.8164 \pm 0.021$ |
| GigaPath + VarMIL | $0.8274 \pm 0.019$ | $0.8689 \pm 0.018$ | $0.7691 \pm 0.018$ | $0.8192 \pm 0.020$ |
| GigaPath + ABMIL | $0.8294 \pm 0.019$ | $0.8712 \pm 0.017$ | $0.7711 \pm 0.019$ | $0.8205 \pm 0.019$ |
| H-Optimus-0 + DSMIL | $0.8365 \pm 0.020$ | $0.8731 \pm 0.018$ | $0.7694 \pm 0.019$ | $0.8237 \pm 0.021$ |
| H-Optimus-0 + CLAM | $0.8373 \pm 0.019$ | $0.8752 \pm 0.017$ | $0.7713 \pm 0.018$ | $0.8253 \pm 0.020$ |
| H-Optimus-0 + VarMIL | $0.8401 \pm 0.018$ | $0.8774 \pm 0.016$ | $0.7735 \pm 0.018$ | $0.8282 \pm 0.019$ |
| H-Optimus-0 + ABMIL | $0.8412 \pm 0.018$ | $0.8786 \pm 0.017$ | $0.7742 \pm 0.017$ | $0.8295 \pm 0.018$ |
| *Fine-tuned (FT) PFM with MIL Aggregators* | | | | |
| UNI + DSMIL (FT) | $0.8132 \pm 0.023$ | $0.8209 \pm 0.021$ | $0.7526 \pm 0.022$ | $0.7837 \pm 0.024$ |
| UNI + CLAM (FT) | $0.8147 \pm 0.022$ | $0.8223 \pm 0.020$ | $0.7542 \pm 0.021$ | $0.7865 \pm 0.023$ |
| UNI + VarMIL (FT) | $0.8176 \pm 0.021$ | $0.8252 \pm 0.020$ | $0.7568 \pm 0.020$ | $0.7894 \pm 0.022$ |
| UNI + ABMIL (FT) | $0.8193 \pm 0.021$ | $0.8237 \pm 0.021$ | $0.7581 \pm 0.021$ | $0.7922 \pm 0.021$ |
| GigaPath + DSMIL (FT) | $0.8372 \pm 0.020$ | $0.8831 \pm 0.017$ | $0.7986 \pm 0.019$ | $0.8351 \pm 0.020$ |
| GigaPath + CLAM (FT) | $0.8381 \pm 0.019$ | $0.8847 \pm 0.016$ | $0.8012 \pm 0.018$ | $0.8377 \pm 0.019$ |
| GigaPath + VarMIL (FT) | $0.8407 \pm 0.018$ | $0.8873 \pm 0.015$ | $0.8043 \pm 0.017$ | $0.8408 \pm 0.018$ |
| GigaPath + ABMIL (FT) | $0.8393 \pm 0.019$ | $0.8858 \pm 0.016$ | $0.8074 \pm 0.018$ | $0.8421 \pm 0.017$ |
| H-Optimus-0 + DSMIL (FT) | $0.8478 \pm 0.019$ | $0.8843 \pm 0.016$ | $0.8121 \pm 0.018$ | $0.8479 \pm 0.019$ |
| H-Optimus-0 + CLAM (FT) | $0.8491 \pm 0.018$ | $0.8859 \pm 0.015$ | $0.8143 \pm 0.017$ | $0.8512 \pm 0.018$ |
| H-Optimus-0 + VarMIL (FT) | $0.8526 \pm 0.017$ | $0.8889 \pm 0.014$ | $0.8167 \pm 0.016$ | $0.8543 \pm 0.017$ |
| H-Optimus-0 + ABMIL (FT) | $0.8512 \pm 0.018$ | $0.8874 \pm 0.015$ | $0.8189 \pm 0.017$ | $0.8529 \pm 0.016$ |
| *Proposed TAPFM method* | | | | |
| UNI (TAPFM) | $0.8415 \pm 0.021$ | $0.8536 \pm 0.019$ | $0.8175 \pm 0.018$ | $0.8309 \pm 0.020$ |
| Gigapath (TAPFM) | $\underline{0.8624 \pm 0.016}$ | $\underline{0.8881 \pm 0.013}$ | $\underline{0.8376 \pm 0.015}$ | $\underline{0.8482 \pm 0.009}$ |
| H-Optimus-0 (TAPFM) | $\mathbf{0.8837 \pm 0.018}$ | $\mathbf{0.8956 \pm 0.011}$ | $\mathbf{0.8503 \pm 0.013}$ | $\mathbf{0.8590 \pm 0.010}$ |

Table 2: Performance comparison on multi-label classification of actionable mutations in LUAD. Results are reported as AUC (mean $\pm$95% CI from 5-fold cross-validation). Based on binary classification results (Table 1), only the top performers are shown, as performance varied primarily by foundation model type.

| Model | **Institutional Cohort** ($N = 876$) | | | | **Macro Average** |
|---|---|---|---|---|---|
| | *EGFR* | *KRAS* | *MET* | *ALK* | |
| UNI (TAPFM) | $0.8287 \pm 0.019$ | $0.7814 \pm 0.021$ | $0.8095 \pm 0.028$ | $0.8341 \pm 0.031$ | $0.8134 \pm 0.018$ |
| Gigapath (TAPFM) | $0.8595 \pm 0.015$ | $0.8075 \pm 0.018$ | $0.8350 \pm 0.024$ | $0.8632 \pm 0.026$ | $0.8413 \pm 0.014$ |
| H-Optimus-0 (TAPFM) | $\mathbf{0.8665 \pm 0.018}$ | $\mathbf{0.8153 \pm 0.019}$ | $\mathbf{0.8420 \pm 0.025}$ | $\mathbf{0.8702 \pm 0.024}$ | $\mathbf{0.8485 \pm 0.017}$ |
| **TCGA** ($N = 438$) | | | | | |
| UNI (TAPFM) | $0.8342 \pm 0.022$ | $0.7896 \pm 0.024$ | $0.7913 \pm 0.035$ | $0.8287 \pm 0.038$ | $0.8110 \pm 0.021$ |
| GigaPath (TAPFM) | $0.8603 \pm 0.018$ | $0.8147 \pm 0.020$ | $0.8179 \pm 0.029$ | $0.8540 \pm 0.032$ | $0.8367 \pm 0.021$ |
| H-Optimus-0 (TAPFM) | $\mathbf{0.8629 \pm 0.017}$ | $\mathbf{0.8226 \pm 0.019}$ | $\mathbf{0.8241 \pm 0.030}$ | $\mathbf{0.8563 \pm 0.034}$ | $\mathbf{0.8415 \pm 0.016}$ |

Although convergence behavior remained stable across this range, increasing $\lambda$ produced steeper decay and lower final training loss, with the best performance observed at $\lambda = 1.0$. These results suggest that stronger weighting of the TAL attention supervision improves training efficiency without compromising stability.

**Number of tiles**: Figure 3c shows that increasing the number of tiles per WSI consistently improved the performance of H-Optimus-0 (TAPFM) model (raw loss curves in Appendix 4b). Another exponential decay model, $\log \mathcal{L}_k = a + b\,k$, was fit to training loss curves over epochs $k = 2$ to 20, with estimated convergence rates of $b = -0.0532$ ($R^2 = 0.9731$), $-0.0569$ ($R^2 = 0.9811$), and $-0.0725$ ($R^2 = 0.9752$) for models trained with 25, 50, and 75 tiles per WSI, respectively. The 75-tile model converged by epoch 7 with a validation AUC of 0.8960, while the 50-tile model reached 0.8764 by the same epoch. The 25-tile variant (TAPFM H-Optimus-0) converged earlier—by epoch 5—but plateaued at a lower AUC of 0.8527. These findings indicate that denser tile sampling both accelerates convergence and improves performance on the institutional validation set.

Experimental analysis of incorporating a cosine regularization term (see Appendix A.5) for feature alignment loss in equation 7 revealed no significant variation in *FGFR3* prediction performance. Consequently, $\lambda = 1.0$ and the maximum number of tiles per WSI per epoch that could be accommodated on a single GPU for each PFM are used in all LUAD experiments

## 5 Clinical Impact

Because LUAD harbors a large number of drug-targetable mutations, molecular diagnostics are especially valuable for guiding its targeted therapies [41, 42]. LUAD was the first tumor type to develop official guidelines recommending universal *EGFR* and *ALK* testing [43], which were subsequently expanded in 2018 to include additional actionable genes when sufficient tissue is available [44]. One of the principal challenges remains obtaining adequate DNA from small core biopsies to perform this extensive molecular workup [45]. Although this paradigm has been embraced in LUAD, other malignancies such as BLCA have been sequenced far less frequently outside major centers despite the availability of *FGFR3*-directed therapies [46]. The ability to predict actionable mutations directly from H&E-stained WSIs promises to streamline and economize current diagnostic workflows. By triaging or potentially obviating the need for costly and tissue-requiring molecular assays, TAPFM can help overcome the infrastructure and financial barriers that preclude precision oncology in resource-limited settings. Even when resources are available, TAPFM offers inference that can be performed as soon as the digital image is scanned. Therefore, clinicians can receive molecularly informed results within hours instead of days/weeks. This study demonstrates that a single model (TAPFM) can simultaneously detect multiple targetable mutations in LUAD, and that its performance generalizes from the originating institution to an independent TCGA cohort.

## 6 Conclusion

This work introduces TAPFM, a novel approach for adapting PFMs to specific clinical tasks by leveraging ViT's attention mechanism for MIL aggregation and a detached dual-gradient approach for updating PFM parameters on a single GPU. TAPFM bridges the gap between self-supervised pretraining and supervised downstream adaptation in computational pathology, enabling more effective use of PFMs for clinical applications. The experimental results establish its effectiveness for clinically relevant mutation prediction tasks for BLCA (*FGFR3*) and LUAD (*EGFR*, *KRAS*, *MET*, *ALK*) patients while maintaining computational efficiency. Notably, TAPFM successfully tackles the challenging task of simultaneous prediction of four actionable mutations in LUAD patients, maintaining reasonable performance even for rare mutations like *MET* and *ALK*. Despite promising results, certain limitations of this work can be addressed in future studies. The approach shows potential for extension to additional clinical endpoints including survival analysis, recurrence prediction, and treatment response estimation. Investigations into which specific transformer layers benefit most from task adaptation could optimize the approach by selectively updating only those parameters. Additional external validation across multi-institutional cohorts with diverse scanning protocols and expanded biomarker panels would strengthen the clinical utility of the proposed approach. Implementations that scale to distributed training across multiple GPUs to increase the number of tiles processed per WSI during training may enhance TAPFM's generalization performance.

## Acknowledgments

We thank the research funding provided by a Cancer Center Support Grant from the NIH/NCI (grant number P30CA008748) and the Warren Alpert Foundation through the Warren Alpert Center for

Digital and Computational Pathology at Memorial Sloan Kettering Cancer Center. We also thank Swaraj Nanda, Siddharth Singi, Jamal Benhamida, David Kim, Jie-Fu Chen, Amir Momeni-Boroujeni, Gregory M. Goldgof, and Gabriele Campanella for their contributions to data curation, experiment design, and manuscript preparation.

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

# A    Theoretical Analysis Details

## A.1    Permutation Invariance

A crucial property for MIL models is permutation invariance, which ensures that the order of instances in a bag does not affect the final prediction and below it is shown that the proposed TAPFM methods hold the permutation invariance property.

**Theorem 1** (Permutation Invariance). *Let $X = \{x_1, x_2, ..., x_K\}$ be a bag of instances and $\pi$ be a permutation of indices $\{1, 2, ..., K\}$. The bag representation $Z$ computed by TAPFM is invariant to permutation, i.e., $Z(X) = Z(\{x_{\pi(1)}, x_{\pi(2)}, \ldots, x_{\pi(K)}\})$.*

*Proof.* Let $X = \{x_1, x_2, ..., x_K\}$ be a bag of instances and $\pi$ be a permutation of indices $\{1, 2, ..., K\}$. Let $X_\pi = \{x_{\pi(1)}, x_{\pi(2)}, ..., x_{\pi(K)}\}$ be the permuted bag.

Using matrix notation, let $\mathbf{Z} = [z_1^T, z_2^T, \ldots, z_K^T]^T \in \mathbb{R}^{K \times D}$ be the feature matrix and $\mathbf{a} = [a_1, a_2, \ldots, a_K]^T \in \mathbb{R}^K$ be the attention weight vector for the original ordering.

Let $\mathbf{P}_\pi \in \mathbb{R}^{K \times K}$ be the permutation matrix corresponding to $\pi$. Then, the feature matrix and attention vector for the permuted bag can be written as:

$$\mathbf{Z}_\pi = \mathbf{P}_\pi \mathbf{Z} \tag{16}$$
$$\mathbf{a}_\pi = \mathbf{P}_\pi \mathbf{a} \tag{17}$$

The bag representation for the original ordering is:

$$Z(X) = \mathbf{Z}^T \mathbf{a} = \sum_{i=1}^{K} a_i z_i \tag{18}$$

The bag representation for the permuted ordering is:

$$Z(X_\pi) = \mathbf{Z}_\pi^T \mathbf{a}_\pi \tag{19}$$
$$= (\mathbf{P}_\pi \mathbf{Z})^T (\mathbf{P}_\pi \mathbf{a}) \tag{20}$$
$$= \mathbf{Z}^T \mathbf{P}_\pi^T \mathbf{P}_\pi \mathbf{a} \tag{21}$$
$$= \mathbf{Z}^T \mathbf{a} \tag{22}$$
$$= Z(X) \tag{23}$$

The equality $\mathbf{P}_\pi^T \mathbf{P}_\pi = \mathbf{I}$ holds because permutation matrices are orthogonal. Therefore, $Z(X) = Z(X_\pi)$, which proves that the bag representation is permutation invariant.

In component form, this is equivalent to:

$$Z(X_\pi) = \sum_{i=1}^{K} a_{\pi(i)} z_{\pi(i)} = \sum_{j=1}^{K} a_j z_j = Z(X) \tag{24}$$

where the change of variables $j = \pi^{-1}(i)$ is used. $\qquad\square$

## A.2    Computational Complexity

**Theorem 2** (Computational Complexity). *The computational complexity of TAPFM for processing a single bag of $K$ instances is $\mathcal{O}(K \cdot C_{ViT})$, where $C_{ViT}$ is the complexity of a forward and backward pass through the specific ViT of a PFM for a single instance.*

*Proof.* The computational complexity of TAPFM is analyzed by examining each step of the proposed Algorithm 1 in detail:

- **Feature Extraction (Forward Pass)**:

Each of the $K$ tiles in the bag is passed through the ViT model to extract features. The complexity of processing a single tile depends on the ViT architecture:

- Patch embedding: $\mathcal{O}(P^2 \cdot C \cdot D)$, where $P$ is the patch size, $C$ is the number of channels, and $D$ is the embedding dimension. - Self-attention layers: $\mathcal{O}(L \cdot N^2 \cdot D)$, where $L$ is the number of layers, $N$ is the number of tokens (patches). - MLP blocks: $\mathcal{O}(L \cdot N \cdot D^2)$.

The total complexity for a single tile forward pass is $\mathcal{O}(C_{\text{ViT\_forward}}) = \mathcal{O}(P^2 \cdot C \cdot D + L \cdot N^2 \cdot D + L \cdot N \cdot D^2)$.

For $K$ tiles, the total forward pass complexity is $\mathcal{O}(K \cdot C_{\text{ViT\_forward}})$.

- **Attention Weight Computation**:

  For each tile $i$, the average attention weight is computed by aggregating over $H$ attention heads and $N$ tokens:

  $$a_i = \frac{1}{H} \sum_{h=1}^{H} \frac{1}{N} \sum_{j=1}^{N} A_{cls,j}^h \tag{25}$$

  This requires $\mathcal{O}(H \cdot N)$ operations per tile, resulting in a total complexity of $\mathcal{O}(K \cdot H \cdot N)$ for all tiles.

- **Bag Representation Computation**:

  Computing the weighted sum of feature vectors:

  $$Z = \mathbf{Z}^T \mathbf{a} = \sum_{i=1}^{K} a_i z_i \tag{26}$$

  This requires $\mathcal{O}(K \cdot D)$ operations, where $D$ is the feature dimension.

- **Aggregator Forward and Backward Pass**:

  The aggregator applies a linear transformation followed by a sigmoid activation:

  $$\hat{y} = \sigma(WZ + b) \tag{27}$$

  The forward pass has complexity $\mathcal{O}(D)$ for the matrix-vector multiplication.

  Computing the aggregator loss and its gradient with respect to the bag representation has complexity $\mathcal{O}(D)$.

  Updating the aggregator parameters has complexity $\mathcal{O}(D)$.

- **Computing Gradients for the PFM**:

  The feature loss is:

  $$\mathcal{L}_{feature} = -\text{tr}(\mathbf{Z}\mathbf{G}_z^T) = \sum_{i=1}^{K} \langle z_i, g_{z_i} \rangle \tag{28}$$

  Computing this loss has complexity $\mathcal{O}(K \cdot D)$.

  The attention loss is:

  $$\mathcal{L}_{attention} = \mathbf{a}^T \mathbf{g}_a = \sum_{i=1}^{K} a_i \cdot g_{a_i} \tag{29}$$

  Computing this loss has complexity $\mathcal{O}(K)$.

- **PFM Backward Pass**:

  The backward pass through the ViT for all $K$ tiles has a complexity of $\mathcal{O}(K \cdot C_{\text{ViT\_backward}})$, which is typically on the same order as the forward pass: $\mathcal{O}(C_{\text{ViT\_backward}}) = \mathcal{O}(P^2 \cdot C \cdot D + L \cdot N^2 \cdot D + L \cdot N \cdot D^2)$.

Combining all steps, the total computational complexity is:

$$\mathcal{O}(K \cdot C_{\text{ViT\_forward}} + K \cdot H \cdot N + K \cdot D + D + K \cdot D + K + K \cdot C_{\text{ViT\_backward}}) \quad (30)$$

Since $C_{\text{ViT}} = C_{\text{ViT\_forward}} + C_{\text{ViT\_backward}}$ and typically $C_{\text{ViT}} \gg H \cdot N$ and $C_{\text{ViT}} \gg D$, the overall complexity is dominated by the ViT forward and backward passes, resulting in $\mathcal{O}(K \cdot C_{\text{ViT}})$.

This complexity analysis shows that the proposed TAPFM approach scales linearly with the number of instances in the bag. The constant factor $C_{\text{ViT}}$ depends on the specific architecture of the Vision Transformer but is independent of the bag size. $\square$

### A.3 Space Complexity Analysis

In addition to time complexity, space complexity is a critical consideration for practical deployment of TAPFM, especially for gigapixel WSIs processed using transformer-based models.

**Theorem 3** (Space Complexity). *The space complexity of TAPFM for processing a single bag of $K$ instances (tiles) is $\mathcal{O}(|\theta_{PFM}| + K \cdot (S_{act} + S_{grad}))$, where $|\theta_{PFM}|$ is the size of the PFM parameters, and $S_{act}$ and $S_{grad}$ are the memory requirements for activations and gradients per instance, respectively.*

*Proof.* The space complexity of TAPFM is analyzed by examining the memory requirements of each component in the proposed approach as follows:

- **Model Parameter Storage**: The memory required to store the ViT parameters is $\mathcal{O}(|\theta_{PFM}|)$, which can be further decomposed as $\mathcal{O}(L \cdot N \cdot D^2)$, where $L$ is the number of transformer layers, $N$ is the number of tokens, and $D$ is the embedding dimension.

- **Feature Matrix Storage**: For a bag of $K$ instances, the feature matrix $\mathbf{Z} \in \mathbb{R}^{K \times D}$ requires $\mathcal{O}(K \cdot D)$ memory.

- **Attention Vector Storage**: The attention weight vector $\mathbf{a} \in \mathbb{R}^K$ requires $\mathcal{O}(K)$ memory.

- **Activation Memory**: During the forward pass, each instance requires storing intermediate activations for all transformer layers. For a single instance, this requires $\mathcal{O}(L \cdot N \cdot D)$ memory for hidden states and $\mathcal{O}(L \cdot H \cdot N^2)$ for attention maps, where $H$ is the number of attention heads. Let's denote the total activation memory per instance as $S_{act} = \mathcal{O}(L \cdot N \cdot D + L \cdot H \cdot N^2)$.

- **Gradient Memory**: During backpropagation, gradients for both activations and parameters must be stored. The memory requirement for gradients per instance is denoted as $S_{grad}$, which is typically on the same order as $S_{act}$.

- **Aggregator Memory**: The aggregator requires $\mathcal{O}(D)$ memory for parameters and $\mathcal{O}(1)$ for the output, which is negligible compared to the PFM memory requirements.

- **Gradient Matrices for Feature and Attention**: The feature gradient matrix $\mathbf{G}_z \in \mathbb{R}^{K \times D}$ requires $\mathcal{O}(K \cdot D)$ memory, and the attention gradient vector $\mathbf{g}_a \in \mathbb{R}^K$ requires $\mathcal{O}(K)$ memory.

Combining all components and identifying the dominant terms, the total space complexity is:

$$\mathcal{O}(|\theta_{PFM}| + K \cdot D + K + K \cdot S_{act} + K \cdot S_{grad} + K \cdot D + K) \quad (31)$$

Since $S_{act}$ and $S_{grad}$ are typically much larger than $D$, and combining like terms produces:

$$\mathcal{O}(|\theta_{PFM}| + K \cdot (S_{act} + S_{grad})) \quad (32)$$

Above equation indicates that the memory requirements of TAPFM scale linearly with the number of instances (tiles) $K$. However, the constant factors $S_{act}$ and $S_{grad}$ can be substantial for large ViT models, potentially limiting the number of instances that can be processed simultaneously on a single GPU. $\square$

## A.4 Task-Specific Adaptation without Catastrophic Forgetting

**Proposition 2** (Preservation of Pretrained Knowledge). *The proposed TAPFM approach effectively implements a specialized form of continual learning that prevents catastrophic forgetting. The Fisher information matrix for pretrained parameters is implicitly preserved:*

$$F_{\theta_{PFM}} = \mathbb{E}_{p(x,y)}[\nabla_{\theta_{PFM}} \log p(z|x; \theta_{PFM}) \nabla_{\theta_{PFM}} \log p(z|x; \theta_{PFM})^T] \tag{33}$$

*Proof.* Catastrophic forgetting occurs when fine-tuning a pre-trained model on a new task destroys the knowledge acquired during pretraining. In information-theoretic terms, this happens when parameter updates move away from regions of high Fisher information with respect to the pretraining task.

The Fisher information matrix $F_{\theta_{PFM}}$ characterizes the curvature of the loss landscape around the pretrained parameters. It quantifies how sensitive the model's output is to small changes in each parameter. Parameters with high Fisher information are crucial for the model's performance on the pretraining task.

In standard fine-tuning, the gradient update is:

$$\Delta\theta_{PFM} = -\eta\nabla_{\theta_{PFM}}\mathcal{L}_{task} \tag{34}$$

This update does not account for the importance of parameters for the pretraining task, potentially leading to catastrophic forgetting. In contrast, optimal updates for preserving pretrained knowledge while adapting to a new task would be of the form:

$$\Delta\theta_{PFM} = -\eta F_{\theta_{PFM}}^{-1}\nabla_{\theta_{PFM}}\mathcal{L}_{task} \tag{35}$$

This is a form of natural gradient descent, which adapts the update direction based on the curvature of the loss landscape.

The proposed TAPFM approach implicitly approximates this behavior through gradient detachment. By separating the aggregator and PFM optimization, TAPFM allows the PFM to adapt more conservatively. The detached gradients from the aggregator act as a filtered signal that guides the PFM to adapt while respecting its pretrained structure.

Specifically, the proposed PFM update is:

$$\Delta\theta_{PFM} = -\eta\nabla_{\theta_{PFM}}\mathcal{L}_{PFM} \tag{36}$$

where $\mathcal{L}_{PFM}$ is defined using detached gradients from the aggregator. This approach resembles a regularized optimization problem:

$$\min_{\theta_{PFM}} \mathcal{L}_{task}(\theta_{PFM}) + \lambda \cdot \mathcal{R}(\theta_{PFM} - \theta_{PFM}^{pretrained}) \tag{37}$$

where $\mathcal{R}$ is an implicit regularization term that penalizes deviations from the pretrained parameters. The solution to this regularized problem can be approximated as:

$$\Delta\theta_{PFM} \approx -\eta(I + \lambda \cdot F_{\theta_{PFM}})^{-1}\nabla_{\theta_{PFM}}\mathcal{L}_{task} \tag{38}$$

As $\lambda$ increases, this approaches the natural gradient update:

$$\Delta\theta_{PFM} \approx -\eta\lambda^{-1}F_{\theta_{PFM}}^{-1}\nabla_{\theta_{PFM}}\mathcal{L}_{task} \approx -\eta' F_{\theta_{PFM}}^{-1}\nabla_{\theta_{PFM}}\mathcal{L}_{task} \tag{39}$$

where $\eta' = \eta\lambda^{-1}$ is an effective learning rate.

The detached gradient approach of TAPFM serves as an implicit regularization mechanism that enables task-specific adaptation while preserving the foundational knowledge embedded in the pretrained model. This balance between adaptation and preservation is especially important for PFMs, where pretraining captures general tissue morphology and fine-tuning adapts to specific clinically relevant tasks. □

### A.5 Cosine Regularization for Feature Alignment Loss

The feature alignment loss in equation 5 encourages feature vectors to move in the direction that reduces the classification loss. However, in certain scenarios, direct optimization of this loss may lead to updates where feature vectors and gradients point in opposing directions, potentially causing unstable training dynamics. To mitigate this issue, a cosine regularization term is considered. The cosine regularization term is defined as: $\mathcal{L}_{reg} = \sum_{i=1}^{K}(1 - \cos(z_i, g_{z_i}))$ where $\cos(z_i, g_{z_i})$ represents the cosine similarity between the feature vector $z_i$ and its gradient $g_{z_i}$, calculated as:

$$\cos(z_i, g_{z_i}) = \frac{\langle z_i, g_{z_i} \rangle}{\|z_i\|_2 \cdot \|g_{z_i}\|_2} \tag{40}$$

This regularization term penalizes scenarios where feature vectors and their gradients are oriented in opposite directions (negative cosine similarity). When incorporated into the task adaptation loss, the complete objective becomes:

$$\mathcal{L}_{PFM} = \mathcal{L}_{feature} + \lambda \mathcal{L}_{attention} + \beta \mathcal{L}_{reg} \tag{41}$$

where $\beta$ is a hyperparameter controlling the strength of the regularization. The purpose of this regularization is to promote more stable optimization by encouraging gradual updates to the learned representations while still allowing adaptation to the downstream task. Theoretically, this prevents drastic changes in feature space that might compromise the pretrained knowledge while ensuring that updates remain aligned with the classification objective.

## B Training Loss Curves

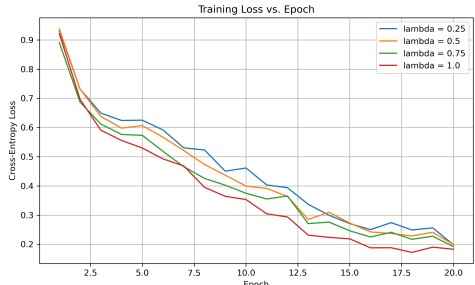

(a) Training loss variation with respect to lambda in TAL (equation 7) for H-Optimus-0 (TAPFM) model.

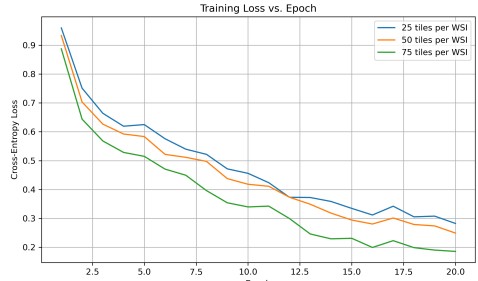

(b) Training loss variation with the number of tiles for H-Optimus-0 (TAPFM) model.

Figure 4: Ablations studies for *FGFR3* prediction in BLCA patients.

## C TAPFM Optimization Stability

Figure 5 empirically demonstrates the stabilization benefit of TAPFM's detached gradient approach compared to unified computational graph training. When attempting joint optimization with a unified graph (left panel), the training loss exhibits unstable oscillations around 0.85 without convergence over 20 epochs, indicating optimization conflicts between the PFM and aggregator parameters. In contrast, TAPFM's detached gradient approach (right panel) shows smooth exponential decay from 0.89 to below 0.08, confirming the theoretical analysis in Section 3.4. This stable convergence enables effective single-GPU training where joint optimization fails due to competing gradient signals through shared parameters.

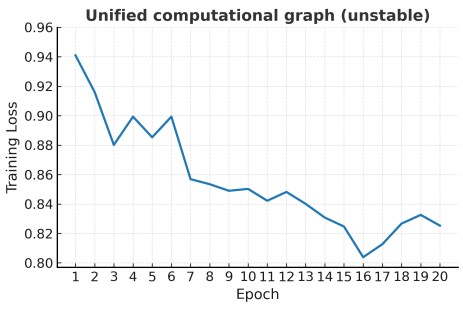
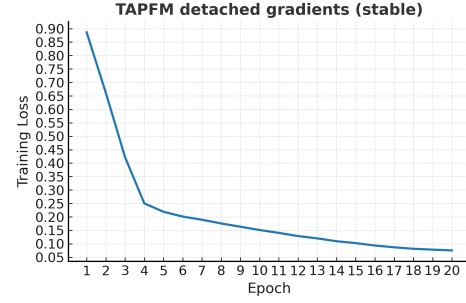

(a) Unified computational graph (unstable)  (b) TAPFM detached gradients (stable)

Figure 5: Training loss comparison demonstrating optimization stability. Unified computational graph training fails to converge due to gradient conflicts, while TAPFM's detached approach enables smooth convergence.

# D    Additional Results

## D.1    Mutation Prediction

To provide a comprehensive assessment accounting for class imbalance, we report Area Under the Precision-Recall Curve (AUPRC) values for the mutation prediction experiments. Tables 3 and 4 present AUPRC results for the binary and multi-label mutation prediction tasks, respectively.

## D.2    Survival Prediction

To demonstrate TAPFM's versatility beyond mutation prediction, we evaluated its performance on overall survival prediction using TCGA-BLCA and TCGA-LUAD cohorts. For this task, we replaced the classification head with a multi-task logistic regression (MTLR) layer and used partial log-likelihood loss for optimization [37]. Results in Table 5 demonstrate that TAPFM's task adaptation approach extends effectively to time-to-event prediction tasks as well.

Table 3: Performance comparison of different PFM and MIL aggregation methods for binary classification tasks in terms of AUPRC (mean $\pm$ 95% CI from 5-fold cross-validation). N indicates the number of patients in the testing cohort. Best and second best models are in bold and underlined text, respectively.

| Model | BLCA *FGFR3* | | LUAD *EGFR* | |
| | Institutional (N = 194) | TCGA (N = 260) | Institutional (N = 876) | TCGA (N = 438) |
|---|---|---|---|---|
| *Fixed-PFM with Trained MIL Aggregators* | | | | |
| UNI + DSMIL | $0.4523 \pm 0.047$ | $0.4687 \pm 0.048$ | $0.4785 \pm 0.038$ | $0.5142 \pm 0.042$ |
| UNI + CLAM | $0.4571 \pm 0.046$ | $0.4712 \pm 0.047$ | $0.4831 \pm 0.037$ | $0.5189 \pm 0.041$ |
| UNI + VarMIL | $0.4618 \pm 0.045$ | $0.4763 \pm 0.046$ | $0.4893 \pm 0.036$ | $0.5247 \pm 0.040$ |
| UNI + ABMIL | $0.4597 \pm 0.046$ | $0.4789 \pm 0.046$ | $0.4862 \pm 0.037$ | $0.5312 \pm 0.039$ |
| GigaPath + DSMIL | $0.5634 \pm 0.041$ | $0.6247 \pm 0.039$ | $0.5473 \pm 0.034$ | $0.6189 \pm 0.037$ |
| GigaPath + CLAM | $0.5683 \pm 0.040$ | $0.6294 \pm 0.038$ | $0.5521 \pm 0.033$ | $0.6235 \pm 0.036$ |
| GigaPath + VarMIL | $0.5741 \pm 0.039$ | $0.6352 \pm 0.037$ | $0.5584 \pm 0.032$ | $0.6298 \pm 0.035$ |
| GigaPath + ABMIL | $0.5789 \pm 0.039$ | $0.6397 \pm 0.036$ | $0.5627 \pm 0.032$ | $0.6341 \pm 0.034$ |
| H-Optimus-0 + DSMIL | $0.5891 \pm 0.038$ | $0.6453 \pm 0.036$ | $0.5692 \pm 0.031$ | $0.6429 \pm 0.036$ |
| H-Optimus-0 + CLAM | $0.5924 \pm 0.038$ | $0.6512 \pm 0.035$ | $0.5748 \pm 0.030$ | $0.6481 \pm 0.035$ |
| H-Optimus-0 + VarMIL | $0.5987 \pm 0.037$ | $0.6583 \pm 0.034$ | $0.5809 \pm 0.029$ | $0.6547 \pm 0.034$ |
| H-Optimus-0 + ABMIL | $0.6023 \pm 0.037$ | $0.6621 \pm 0.034$ | $0.5841 \pm 0.029$ | $0.6583 \pm 0.033$ |
| *Fine-tuned (FT) PFM with MIL Aggregators* | | | | |
| UNI + DSMIL (FT) | $0.5247 \pm 0.044$ | $0.5389 \pm 0.045$ | $0.5134 \pm 0.036$ | $0.5497 \pm 0.040$ |
| UNI + CLAM (FT) | $0.5293 \pm 0.043$ | $0.5427 \pm 0.044$ | $0.5189 \pm 0.035$ | $0.5558 \pm 0.039$ |
| UNI + VarMIL (FT) | $0.5361 \pm 0.042$ | $0.5487 \pm 0.043$ | $0.5258 \pm 0.034$ | $0.5631 \pm 0.038$ |
| UNI + ABMIL (FT) | $0.5412 \pm 0.042$ | $0.5463 \pm 0.044$ | $0.5297 \pm 0.034$ | $0.5689 \pm 0.037$ |
| GigaPath + DSMIL (FT) | $0.5923 \pm 0.039$ | $0.6687 \pm 0.034$ | $0.6142 \pm 0.031$ | $0.6547 \pm 0.035$ |
| GigaPath + CLAM (FT) | $0.5971 \pm 0.038$ | $0.6734 \pm 0.033$ | $0.6213 \pm 0.030$ | $0.6608 \pm 0.034$ |
| GigaPath + VarMIL (FT) | $0.6047 \pm 0.037$ | $0.6812 \pm 0.032$ | $0.6301 \pm 0.029$ | $0.6689 \pm 0.033$ |
| GigaPath + ABMIL (FT) | $0.6021 \pm 0.038$ | $0.6781 \pm 0.033$ | $0.6358 \pm 0.029$ | $0.6723 \pm 0.032$ |
| H-Optimus-0 + DSMIL (FT) | $0.6187 \pm 0.036$ | $0.6794 \pm 0.033$ | $0.6473 \pm 0.028$ | $0.6831 \pm 0.034$ |
| H-Optimus-0 + CLAM (FT) | $0.6234 \pm 0.035$ | $0.6847 \pm 0.032$ | $0.6541 \pm 0.027$ | $0.6912 \pm 0.033$ |
| H-Optimus-0 + VarMIL (FT) | $0.6318 \pm 0.034$ | $0.6931 \pm 0.031$ | $0.6624 \pm 0.026$ | $0.6989 \pm 0.032$ |
| H-Optimus-0 + ABMIL (FT) | $0.6289 \pm 0.035$ | $0.6897 \pm 0.032$ | $0.6671 \pm 0.027$ | $0.6958 \pm 0.031$ |
| *Proposed TAPFM method* | | | | |
| UNI (TAPFM) | $0.6142 \pm 0.041$ | $0.6389 \pm 0.039$ | $0.6542 \pm 0.030$ | $0.6647 \pm 0.036$ |
| Gigapath (TAPFM) | $\mathbf{0.7235 \pm 0.032}$ | $\mathbf{0.7251 \pm 0.028}$ | $0.7089 \pm 0.025$ | $0.7143 \pm 0.021$ |
| H-Optimus-0 (TAPFM) | $\underline{0.7163 \pm 0.035}$ | $\underline{0.7164 \pm 0.024}$ | $\mathbf{0.7287 \pm 0.023}$ | $\mathbf{0.7319 \pm 0.017}$ |

Table 4: Performance comparison on multi-label classification of actionable mutations in LUAD in terms of AUPRC (mean $\pm$ 95% CI from 5-fold cross-validation). Based on binary classification results (Table 1), only the top performers are shown, as performance varied primarily by foundation model type.

| Model | Institutional Cohort ($N = 876$) | | | | Macro Average |
| | *EGFR* | *KRAS* | *MET* | *ALK* | |
|---|---|---|---|---|---|
| UNI (TAPFM) | $0.6589 \pm 0.032$ | $0.6047 \pm 0.036$ | $0.1673 \pm 0.061$ | $0.1429 \pm 0.068$ | $0.3934 \pm 0.031$ |
| Gigapath (TAPFM) | $\mathbf{0.6872 \pm 0.027}$ | $\mathbf{0.6396 \pm 0.031}$ | $0.2483 \pm 0.055$ | $0.1507 \pm 0.062$ | $0.4315 \pm 0.026$ |
| H-Optimus-0 (TAPFM) | $0.6773 \pm 0.025$ | $0.6177 \pm 0.029$ | $\mathbf{0.3457 \pm 0.051}$ | $\mathbf{0.1951 \pm 0.058}$ | $\mathbf{0.4590 \pm 0.024}$ |
| **TCGA** ($N = 438$) | | | | | |
| UNI (TAPFM) | $0.6631 \pm 0.038$ | $0.6095 \pm 0.042$ | $0.1547 \pm 0.072$ | $0.1392 \pm 0.081$ | $0.3916 \pm 0.036$ |
| Gigapath (TAPFM) | $0.6850 \pm 0.032$ | $\mathbf{0.6413 \pm 0.036}$ | $0.2374 \pm 0.063$ | $0.1431 \pm 0.074$ | $0.4269 \pm 0.031$ |
| H-Optimus-0 (TAPFM) | $\mathbf{0.6942 \pm 0.030}$ | $0.6284 \pm 0.034$ | $\mathbf{0.3104 \pm 0.059}$ | $\mathbf{0.1862 \pm 0.069}$ | $\mathbf{0.4548 \pm 0.029}$ |

Table 5: Overall survival prediction performance comparison on TCGA cohorts. Results are reported as concordance index (C-index) with 95% confidence intervals from 5-fold cross-validation. Higher values indicate better survival prediction.

| method | TCGA-BLCA | | TCGA-LUAD | |
|---|---|---|---|---|
| | C-index | 95% CI | C-index | 95% CI |
| UNI (TAPFM) | 0.6218 | (0.5759, 0.6677) | 0.6687 | (0.6042, 0.7332) |
| GigaPath (TAPFM) | 0.6341 | (0.5935, 0.6747) | 0.6825 | (0.6214, 0.7436) |
| H-Optimus-0 (TAPFM) | **0.6408** | **(0.6045, 0.6771)** | **0.6953** | **(0.6371, 0.7535)** |

