# OpenReview forum: "Single GPU Task Adaptation of Pathology Foundation Models for Whole Slide Image Analysis"
_NeurIPS.cc/2025/Conference — NeurIPS 2025 poster_

### Official Review · Reviewer_JNqh · 2025-06-27

**Clarity:** 3
**Significance:** 2
**Originality:** 2
**Rating:** 4
**Confidence:** 3

**Summary:**

This paper addresses the classification of whole slide pathology images. It proposes a method for adapting pretrained pathology foundation models (PFMs) to specific downstream tasks by decoupling and alternating the optimization of feature representations and attention weights. Experimental evaluations on bladder and lung cancer datasets, for both binary malignancy classification and multi-label mutation prediction, demonstrate improved performance over existing approaches.

**Questions:**

1) Improve the clarity and precision of the technical contributions described in Section 3.
2) Include additional evaluation metrics beyond AUC for the existing experiments on BLCA and LUAD. This would provide a better picture of model performance, capturing not only statistical power but also the utility of predictions on test data. Metrics such as accuracy, precision, recall, F1 score, or calibration measures could demonstrate the robustness and clinical relevance of the method compared to other approaches.
3) Additional minor issues mentioned above.

**Ethical Concerns:**

["NO or VERY MINOR ethics concerns only"]

**Final Justification:**

I’m leaning toward acceptance, though I don’t feel strongly either way. The paper reads well, but it’s not particularly clear on the advantages of the proposed method (aside from the 24×H100 vs. 1×H100 comparison mentioned only in the rebuttal), why it performs well, or shows results that are especially strong. The authors have addressed my comments and questions, but I wasn’t convinced to raise my score.

**Limitations:**

Yes.

**Paper Formatting Concerns:**

Seems okay.

**Quality:**

3

**Strengths And Weaknesses:**

*Presentation and Technical Contributions:*
The paper is generally well written and clearly organized. The proposed approach appears reproducible, and the authors provide code to support replication. However, the technical description in Section 3 lacks clarity. Concretely, the core contribution seems to boil down to disabling gradients and alternating between two optimization steps. While this may still be a reasonable and useful strategy, its novelty and significance are somewhat limited and would benefit from a clearer, more direct, exposition. The justification provided in Proposition 1 feels superficial and does not convincingly support the conclusion that decoupling leads to “more stable parameter trajectories” than joint optimization.
Lastly, the title emphasizes “single GPU,” but the manuscript does not clearly explain whether existing approaches are incompatible with single-GPU training, or whether this limitation is a motivation for the proposed method. It remains unclear if the method is specifically designed to reduce computational requirements, or if the focus is instead on improving training dynamics and predictive performance. Clarifying this point would help align the title with the actual contributions and goals of the paper.

*Experimental Evaluation:*
The experimental results are promising and demonstrate improvements over baseline methods. However, the scope of the evaluation is somewhat limited:
(a) It is confined to two tasks (bladder and lung cancer), both using what appears to be a very similar setup (e.g., the same set of classes for multi-label classification).
(b) Some experimental details are unclear; for example, how the locally collected data are integrated with publicly available datasets.
(c) Relying solely on AUC as the evaluation metric reduces the clinical relevance and impact, as it does not capture all aspects of model performance in comparison to other methods.

*Additional Comments:*
* Line 12: Define "TCGA" at first mention.
* Line 13: Define "H-Optimus-0" at first mention.
* Lines 21–24: Please indicate the resolution of a whole-slide image (WSI), in addition to the individual pixel size.
* Line 39: The term "dual-loss mechanism" is unclear and not elaborated on later in the paper. Please clarify.
* Section 2 (especially 2.1): The section is dense with abbreviations, which makes it difficult to follow. Consider simplifying or expanding key terms.
* Line 91: What is a "learnable CLS token"? Please provide an explanation or reference.
* Section 3.2: Please clarify which components (e.g., Equation 1) are based on existing methods and which represent novel contributions.
* Equation 5: Should the notation be $g_{z_i}$ instead?
* Equations 5 and 6: Why are inconsistent notations ($\times$ and $\cdot$) used? Is there a distinction that should be clarified?
* Lines 142–144: Please justify the claim made in this sentence; it currently lacks sufficient support.
* Proposition 1: The statement and proof are vague; please see comments above regarding the need for a clearer and more rigorous justification.
* Line 178: Define "EGFR, KRAS, MET, and ALK" and provide appropriate references.
* Line 189: What is "Otsu thresholding"? A brief explanation or citation would help readers unfamiliar with the term.
* Line 197: Shouldn't the number of tiles used during inference match the number used during training?
* Section 4.2: How does the training/inference runtime compare to that of other (non-decoupled) methods?
* Line 244: Since the loss depends on $\lambda$, does a lower numerical loss value actually reflect better performance?
* Table 2: Why is there no comparison to methods without TAPFM?

---

> ### Author Rebuttal · Authors · 2025-07-31
>
> **Note:** For brevity PMID refers to the PubMed identifier of the papers referenced in the rebuttal. The non-PMID-indexed reference is fully cited. Original reviewer comments are truncated to allow for complete responses.
>
> ### **Reviewer Comment 1**
> *The technical description in Section 3 lacks clarity. … incompatible with single-GPU training.*
>
> **Author Response:**
> We would like to clarify a technical distinction: TAPFM employs detached gradients to strategically direct gradient flow, not to disable it. This is a deliberate design choice to address the instability that arises when fine-tuning pathology foundation models (PFMs) in the MIL setting for gigapixel pathology images. The detachment breaks the computation graph between the PFM and the aggregator module, allowing proper gradient flow during backpropagation (Equations 8–10). This design makes training feasible on a single GPU, where memory constraints prohibit full end-to-end joint optimization of large PFMs with transformer backbones under a MIL framework.
>
> TAPFM also uniquely uses self-attention scores from the ViT backbone to compute slide-level MIL aggregation. This internal use of ViT attention for MIL has not been described before (see lines 95–97).
>
> Conventional approaches freeze the PFM and train only the MIL aggregator, resulting in suboptimal performance (see references 11–13 and Table 1). Campanella et al. (*Nature Medicine*, July 2025, PMID: 40634781) demonstrated that adapting foundation models to mutation prediction substantially improves performance, achieving AUCs of 0.847 (institutional) and 0.860 (TCGA) for *EGFR* prediction using distributed training on 24 H100 GPUs. However, this distributed strategy is unavailable for single-GPU systems where computations cannot be parallelized across devices.
>
> An alternative is encoder-aggregator training in a unified computational graph on a single GPU, but this suffers from optimization conflicts. In the MIL framework, the bag representation $Z =$ **Z**$^T$**a** requires the PFM to simultaneously optimize feature extraction (**Z**) and attention weighting (**a**) through shared parameters $θ_{PFM}$. Gradients from both pathways ($∂Z/∂$**Z** · $∂$**Z**$/∂θ_{PFM}$ and $∂Z/∂$**a** · $∂$**a**$/∂θ_{PFM}$) create conflicting optimization signals. TAPFM's detached-gradient formulation decouples this: aggregator learns from feature / attention combinations using fixed PFM outputs, then PFM adapts without competing aggregator gradients.
>
> This enables fine-tuning PFMs on a single GPU without compromising performance: TAPFM achieves AUCs of 0.867 (institutional) and 0.863 (TCGA), comparable to the 24-GPU baseline. We can include training loss curves comparing unified vs. detached optimization to empirically demonstrate this effect, and will reference Campanella et al.’s work in the final manuscript to provide context for our single-GPU results.
>
>
> ### **Reviewer Comment 2**
> *Evaluation scope is limited:*
>
> >*(a) Confined to two tasks...*
>
> >*(b) Some details are unclear...*
>
> >*(c) Sole reliance on AUC...*
>
> **Author Response:**
> We appreciate the reviewer's thoughtful feedback and address each concern below.
>
> **(a) Evaluation Scope**
> TAPFM's evaluation demonstrates its effectiveness across clinically actionable biomarkers with direct therapeutic implications, as detailed in Section 5. *FGFR3* mutations in bladder cancer guide treatment with erdafitinib. In LUAD, *EGFR*, *ALK*, *KRAS*, and *MET* mutations inform treatment with personalized therapies (e.g. sotorasib and tyrosine kinase inhibitors).
>
> The multi-label framework addresses a fundamental clinical need that conventional classification cannot. As shown in Table 2, TAPFM enables simultaneous prediction of co-occurring mutations. While binary classification handles one mutation at a time and multi-class frameworks assume mutual exclusivity, multi-label classification models concurrent mutations, a reality in LUAD with coexisting *EGFR*, *KRAS*, *MET*, and *ALK* alterations (please see response to cPrU for more details).
>
> **(b) Data Integration Methodology**
> No complex integration is required. Both institutional and TCGA datasets contain WSIs matched with mutation labels derived from molecular testing. Variations in slide preparation and scanning are expected and useful for robust evaluation.
>
> **(c) Evaluation Metrics**
> We will include confidence intervals from 5-fold cross validation (CV) using institutional data and TCGA as independent test sets. We will also report AUPRC values to support performance interpretation in appendix. We avoid threshold-based metrics (e.g., F1, accuracy) due to their sensitivity to cutoffs, which limits their clinical interpretability.
>
> ### **Reviewer Comment 3**
> *TCGA*
>
> **Author Response:**
> We will define TCGA as *The Cancer Genome Atlas* at first mention.
>
> ---
>
> ### **Reviewer Comment 4**
> *"H-Optimus-0"*
>
> **Author Response:**
> We use this as a model name (ref 9).
>
> ---
>
> ### **Reviewer Comment 5**
> *resolution of a whole-slide image*
>
> **Author Response:**
> We will replace “magnification” with “resolution” in line 21 for clarity.
>
> ---
>
> ### **Reviewer Comment 6**
> *"dual-loss mechanism"*
>
> **Author Response:**
> We will clarify that the term refers to the use of two loss function: classification (Equation 4) and task adaptation (Equation 7).
>
> ---
>
> ### **Reviewer Comment 7**
> *Section 2 (especially 2.1) ... Consider simplifying or expanding key terms.*
>
> **Author Response:**
> The names of the algorithms, not abbreviations, as named in the original cited papers are mentioned in Section 2.
>
> ---
>
> ### **Reviewer Comment 8**
> *Line 91: "learnable CLS token"*
>
> **Author Response:**
> We refer to the CLS token from ViT models and have cited the original ViT manuscript in *line 89*.
>
> ---
>
> ### **Reviewer Comment 9**
> *Section 3.2: Please clarify ... which represent novel contributions.*
>
> **Author Response:**
> The following are novel contributions:
>
> a. MIL aggregation using internal attention weights (averaged across heads and tokens) for ViT-based PFMs (lines 79–81, 96–97).
> b. Separate aggregator and encoder optimization in distinct computation graphs on a single GPU (lines 103–104, 115–116).
> c. Scaling of attention weights for use in MIL aggregation (lines 104–109).
> d. Task adaptation loss combining novel feature alignment and attention loss components (Section 3.3).
> e. Theoretical analysis of TAPFM and its properties (Section 3.4, Appendix A.1–A.4).
>
> ---
>
> ### **Reviewer Comment 10**
> *Equation 5: Should the notation be $g_{z_{i}}$ instead?*
>
> **Author Response:**
> Yes, we will correct this.
>
> ---
>
> ### **Reviewer Comment 11**
> *Equations 5 and 6: Why are inconsistent notations ( x and .) used? ...*
>
> **Author Response:**
> Both are scalar multiplications. We will use “x” consistently.
>
> ### **Reviewer Comment 12**
> *Lines 142–144: Please justify the claim made in this sentence; it currently lacks sufficient support.*
>
> **Author Response:**
> Please see response to comment 1.
>
> ### **Reviewer Comment 13**
> *Define "EGFR, KRAS, MET, and ALK" ...*
>
> **Author Response:**
> These are names of common proto-oncogenes that are extensively studied in clinical oncology and molecular pathology. We have provided appropriate clinical references in Section 5 of the manuscript. We will ensure all gene names are italicized to indicate that they refer to genes. The names no longer reflect their original derivations; for example, *KRAS* was originally derived from “Kirsten Rat Sarcoma Viral Oncogene Homolog” but is now formally named “KRAS Proto-Oncogene, GTPase”.
>
> ### **Reviewer Comment 14**
> *"Otsu thresholding"*
>
> **Author Response:**
> We will include the following reference:
>
> > **Otsu, Nobuyuki.**
> > *A Threshold Selection Method from Gray-Level Histograms.*
> > IEEE Transactions on Systems, Man, and Cybernetics, Jan. 1979.
>
> ### **Reviewer Comment 15**
> *... tiles used during inference match the number used during training?*
>
> **Author Response:**
> Training is memory intensive as it involves gradient tracking through massive PFM and MIL aggregator parameters, limiting the number of tiles per slide on a single GPU. Inference, however does not have this constraint (see response to comment 7 for reviewer ggKh). Slide-level predictions require that all tiles be evaluated.
>
> ### **Reviewer Comment 16**
> *How does the training/inference runtime compare to that of other (non-decoupled) methods?*
>
> **Author Response:**
> To the best of our knowledge, there are no decoupled methods for PFM task adaptation for ViT-based PFMs in the MIL setting. Contemporary approaches rely on distributed training across multiple GPUs (lines 77–81), with Campanella et al. (*Nature Medicine*, July 2025, PMID: 40634781) using 24 H100 GPUs.
>
> We identified calculation errors in the training times reported in Section 4.2 and will update them in the revised manuscript. The correct TAPFM training durations are: BLCA – 12h (UNI), 21h (Gigapath), 24h (H-Optimus-0); LUAD – 2d 4h (UNI), 4d 2h (Gigapath), 4d 6h (H-Optimus-0). Average inference time per WSI is 4.85–7.15 minutes (lines 225–226).
>
> ### **Reviewer Comment 16**
> *... lambda, does a lower numerical loss value actually reflect better performance?*
>
> **Author Response:**
> The Task Adaptation Loss (TAL, Equation 7) is designed as a surrogate objective that encourages feature alignment with the classification task. Lower TAL values may correlate with better performance by design and as supported by the ablation results in Figure 2b and Section 4.4.
>
> ### **Reviewer Comment 17**
> *Table 2: Why is there no comparison to methods without TAPFM?*
>
> **Author Response:**
> Please see response to Comment 4 for reviewer cPrU.
>
> ### **Reviewer Comment 18**
> *Include additional evaluation metrics beyond AUC for BLCA and LUAD... Metrics like accuracy, precision, recall, F1, or calibration could show robustness and clinical relevance.*
>
> **Author Response:**
> See response to comment 2(c).

---

> > ### Comment · Reviewer_JNqh · 2025-08-05
> >
> > Thank you for addressing my questions and comments. I will maintain my rating.

---

### Official Review · Reviewer_njqY · 2025-06-30

**Clarity:** 3
**Significance:** 3
**Originality:** 2
**Rating:** 4
**Confidence:** 3

**Summary:**

This paper proposes a method to adapt pathology foundation models for whole slide tissue image (WSI) classification. The proposed method makes use of the attention mechanism of the vision transformer architecture for feature aggregation and employs separate computation graphs for parameter updates.

**Questions:**

Additional investigation is needed to better assess the benefits and limitations of the proposed method. The WSI classification task may not be the best analysis task to evaluate this method. Other analysis tasks such as survival analysis may benefit more.

Moreover, the authors use relatively large institutional datasets to train their models. One of the main benefits of using foundation models is to be able to use relatively limited volume of training data. The authors should consider training their models with TCGA data and use the institutional datasets for testing.

**Ethical Concerns:**

["NO or VERY MINOR ethics concerns only"]

**Final Justification:**

I have updated my rating after reviewing the authors' responses to my questions and comments. The paper has technical merit and contributions but relatively low accuracy improvements over other existing methods, hence, my rating of borderline accept.

**Limitations:**

Some recent works have demonstrated high performances (e.g. [1,2]; [1] shows AUC scores of about 0.97). There should be a comparison with those works as well. While adaptation of PFMs is a reasonable approach, clinical research studies will want to use the best methods available regardless of the underlying methodology.

[1] Dang, Thao M., Yuzhi Guo, Hehuan Ma, Qifeng Zhou, Saiyang Na, Jean Gao, and Junzhou Huang. "MFMF: Multiple Foundation Model Fusion Networks for Whole Slide Image Classification." In Proceedings of the 15th ACM International Conference on Bioinformatics, Computational Biology and Health Informatics, pp. 1-8. 2024.
[2] Li, Hao, Ying Chen, Yifei Chen, Rongshan Yu, Wenxian Yang, Liansheng Wang, Bowen Ding, and Yuchen Han. "Generalizable whole slide image classification with fine-grained visual-semantic interaction." In Proceedings of the IEEE/CVF Conference on Computer Vision and Pattern Recognition, pp. 11398-11407. 2024.

**Quality:**

3

**Strengths And Weaknesses:**

This study introduces a different approach to integrating and adapting foundation models and multi-instance learning (MIL) based classification models for WSI classification tasks. The experiments with several pathology foundation models (PFMs) and MIL approaches show that for certain PFM and MIL integrations, the proposed method achieves good performance improvement (e.g, UNI+DSMIL vs UNI (TAPFM) on TCGA LUAD dataset). However, when the best performance of the proposed method (H-Optimus-0 TAPFM) is compared with that of the best alternative approach (H-Optimus-O+ABMIL FT), the improvements are marginal and may not have much impact in a clinical research setting.

Additional investigation is needed to better assess the benefits and limitations of the proposed method. The WSI classification task may not be the best analysis task to evaluate this method. Other analysis tasks such as survival analysis may benefit more.
Moreover, the authors use relatively large institutional datasets to train their models. One of the main benefits of using foundation models is to be able to use relatively limited volume of training data. The authors should consider training their models with TCGA data and use the institutional datasets for testing.

Some recent works have demonstrated high performances (e.g. [1,2]; [1] shows AUC scores of about 0.97). There should be a comparison with those works as well. While adaptation of PFMs is a reasonable approach, clinical research studies will want to use the best methods available regardless of the underlying methodology.

[1] Dang, Thao M., Yuzhi Guo, Hehuan Ma, Qifeng Zhou, Saiyang Na, Jean Gao, and Junzhou Huang. "MFMF: Multiple Foundation Model Fusion Networks for Whole Slide Image Classification." In Proceedings of the 15th ACM International Conference on Bioinformatics, Computational Biology and Health Informatics, pp. 1-8. 2024.
[2] Li, Hao, Ying Chen, Yifei Chen, Rongshan Yu, Wenxian Yang, Liansheng Wang, Bowen Ding, and Yuchen Han. "Generalizable whole slide image classification with fine-grained visual-semantic interaction." In Proceedings of the IEEE/CVF Conference on Computer Vision and Pattern Recognition, pp. 11398-11407. 2024.

---

> ### Author Rebuttal · Authors · 2025-07-30
>
> **Note:** For brevity PMID refers to the PubMed identifier of the papers referenced in the rebuttal. We have provided full paper details for those articles that do not have a PMID or are not referenced in the manuscript.
>
> ---
>
> ### **Reviewer Comment 1**
> *This paper proposes a method to adapt pathology foundation models for whole slide tissue image (WSI) classification.*
>
> **Author Response:**
> The core TAPFM methodology presented in this paper can be used for various types of downstream pathology tasks including classification, regression, survival prediction, etc. If one replaces the last layer with the task specific layer (such as the Cox proportional hazard layer for survival prediction instead of Equation 3) and uses an appropriate loss function (such as the partial log-likelihood loss for survival prediction instead of Equation 4), the core TAPMF method can be used directly to update the pathology foundation models (PFM) weights to adapt to a variety of clinical tasks. We will mention this additional detail in the final version of the paper as we did not specifically make this point in the original manuscript.
>
> To further assess the utility of TAPFM beyond mutation prediction, we conducted additional experiments on overall survival prediction from whole slide images using the TCGA-BLCA and TCGA-LUAD cohorts. TAPFM achieved concordance indices of **0.6408 ± 0.0185** on BLCA and **0.6953 ± 0.0297** on LUAD, outperforming the previous best results reported in *Tang et al.* (CVPR 2024), which were **0.6398 ± 0.0263** and **0.6719 ± 0.0402**, respectively, on the same datasets.  All of these experiments are done with 5-fold cross validation (CV).
>
> We also note that out-of-the-box foundation model performance has been reported in Figure 4 of *Yang et al.* (*Nature Communications*, April 2025, PMID: 40064883). This figure does not report all confidence intervals but states that their optimized method achieves best performance of **0.6039** (95% CI: 0.5093–0.6986).
>
> Please note that the above survival prediction performances are from image only experiments.  Multimodel survival predictions have been proposed but that is not comparable to our current work.
>
> In the final manuscript, we will report in the appendix a table with our results along with out-of-the-box foundation model performance for the survival prediction task. TCGA is of limited use in real-world scenarios (see response to Comment 5). We chose to highlight tasks that have clinical utility with achievable performance in the main manuscript.
>
> > **Tang, Wenhao, et al.**
> > *Feature re-embedding: Towards foundation model-level performance in computational pathology.*
> > Proceedings of the IEEE/CVF Conference on Computer Vision and Pattern Recognition. 2024.
>
> ---
>
> ### **Reviewer Comment 2**
> *However, when the best performance of the proposed method (H-Optimus-0 TAPFM) is compared with that of the best alternative approach (H-Optimus-O+ABMIL FT), the improvements are marginal and may not have much impact in a clinical research setting.*
>
> **Author Response:**
> We thank the reviewer and respectfully disagree that the observed gains are marginal. H-Optimus-0 TAPFM outperforms H-Optimus-0 + ABMIL FT across all tasks and cohorts (e.g., **0.9021 vs. 0.8874 AUC** for FGFR3, **0.8553 vs. 0.8529 AUC** for EGFR; see Table 1). These differences are clinically meaningful when thresholds are tuned for deployment, as demonstrated in the prospective silent trial *Campanella et al. (Nature Medicine*, July 2025, PMID: 40634781), where a model with comparable AUC reduced EGFR testing by 43%.
>
> Unlike that work, which required 24 Nvidia H100 GPUs, TAPFM achieves this performance using a single GPU. Additionally, TAPFM introduces a novel mechanism leveraging ViT self-attention scores for MIL aggregation and gradient routing, an approach not described in prior literature (see lines 95–97).  We will include reference to *Campanella et al.*'s *Nature Medicine* paper in the final version of the paper to provide additional clinical context.
>
> ---
>
> ### **Reviewer Comment 3**
> *The WSI classification task may not be the best analysis task to evaluate this method. Other analysis tasks such as survival analysis may benefit more.*
>
> **Author Response:**
> It should be noted that not all WSI classification tasks are a monolith.  Different tasks have unique challenges so comparing AUC performance of tumor detection/classification to mutation prediction is inappropriate, see response comment 6.
>
> See response to comment 1 regarding survival analysis.
>
>
> ---
>
> ### **Reviewer Comment 4**
> *Moreover, the authors use relatively large institutional datasets to train their models. One of the main benefits of using foundation models is to be able to use a relatively limited volume of training data. *
>
> **Author Response:**
> While perhaps the first interpretation of the use of PFMs was to demonstrate superior performance to other encoders on small datasets, the limitations of such models have become apparent (see references 11, 12, 29, and 30 in the manuscript), and this is precisely what we intend to address with this work.
>
> Results from Table 1 (for fixed PFMs with trained MIL aggregators) do not yield satisfactory performance for the challenging task of mutation prediction across BLCA and LUAD when the foundation models are used as feature extractors with frozen weights.
>
> Thus, task adaptation of PFMs for clinically relevant mutation prediction becomes necessary and TAPFM provides a method for task adaptation of ViT-based PFMs on a single GPU, which has never been done before to the best of our knowledge.
>
> ---
>
> ### **Reviewer Comment 5**
> *The authors should consider training their models with TCGA data and use the institutional datasets for testing.*
>
> **Author Response:**
> TCGA is a relatively idealized dataset that has large tissue resections that are predominantly tumor as opposed to most material available for sequencing, which are often small biopsies. Further, in the real world, like with our institutional dataset, the tumor slides might have regions of tumor that make up <10% of the tissue area on the slide.
>
> These datasets with lower signals stress test performance and allow for better generalization. Models that are trained exclusively on TCGA are unlikely to generalize to real-world tasks (references 11 and 12 in the manuscript).
>
> As an example, we trained ABMIL models using features obtained from state-of-the-art PFMs on TCGA. When these models were tested on our internal cohort, the mean macro-AUC across PFMs dropped from **0.778 to 0.725** for EGFR prediction in LUAD. In contrast, models trained on our internal cohort and evaluated on TCGA perform better (Table 1, results for fixed PFMs with trained MIL aggregators).
>
> While it is possible to adapt an existing PFM with TAPFM on TCGA data and test on institutional data, the advantages of TAPFM lie in exposing the foundation model to the most diverse dataset available.
>
> ---
>
> ### **Reviewer Comment 6**
> *Some recent works have demonstrated high performances (e.g., [1,2]; [1] shows AUC scores of about 0.97). There should be a comparison with those works as well.*
>
> **[1]** Dang, Thao M., et al. *MFMF: Multiple Foundation Model Fusion Networks for Whole Slide Image Classification.* ACM-BCB 2024.
> **[2]** Li, Hao, et al. *Generalizable whole slide image classification with fine-grained visual-semantic interaction.* CVPR 2024.
>
> **Author Response:**
> The reference works achieved ~0.97 AUC on tumor detection (CAMELYON16) and tumor type classification (TCGA NSCLC: LUAD vs. LUSC) tasks. However, these tasks are fundamentally different from the mutation prediction task addressed in our submission.
>
> Pathologists are trained to visually identify tumors and distinguish between tumor subtypes like LUAD versus LUSC (ref: *WHO Classification of Tumours of the Lung, Pleura, Thymus and Heart*, 4th Ed., 2015), but no visual markers exist for detecting genetic mutations from H&E-stained WSIs.
>
> Moreover, the referenced works focus on building better slide-level classifiers on top of ResNet-based feature extractors. In contrast, modern ViT-based PFMs (refs 5 and 9 in our manuscript) achieve nearly perfect performance on simpler WSI classification tasks right out of the box.
>
> Using the H-Optimus-0 PFM as a fixed feature extractor with ABMIL, we achieved:
>
> - **AUC 0.9824 ± 0.0187** on CAMELYON16
> - **AUC 0.9871 ± 0.0115** on TCGA NSCLC under 5-fold CV
>
> These tasks appear to be effectively “solved” by existing PFMs and do not require adaptation.
>
> In contrast, Table 1 shows that out-of-the-box PFMs perform poorly for clinically relevant mutations (e.g., FGFR3 in BLCA and EGFR in LUAD), as noted in references 11, 12, 29, and 30.
>
> TAPFM adapts ViT-based PFMs to improve performance on these harder tasks. It shows improvements on internal datasets and when run inference-only on TCGA (Table 1).
>
> A recent study by *Campanella et al.* (*Nature Medicine*, July 2025, PMID: 40634781) confirms the benefit of adapting PFMs for EGFR prediction in LUAD. We will include this as a reference in the final version.
>
> That study used 24 Nvidia H100 GPUs to fine-tune Prov-Gigapath, achieving AUC of **0.847** (internal) and **0.860** (TCGA). TAPFM achieves **0.867** (internal) and **0.863** (TCGA) using just one H100 GPU.
>
> TAPFM also supports multi-label classification for co-occurring mutations (Section 3.5, Table 2).
>
> **Clinical Relevance:** NCCN guidelines recommend comprehensive genomic testing for LUAD and BLCA, but these tests are expensive, slow, not universally available, and consume tissue. TAPFM provides a fast, cost-effective, tissue sparing tool as an adjunct to sequencing (see Section 5 of the manuscript).
>
> > **Travis, William D., et al.**
> > *WHO Classification of Tumours of the Lung, Pleura, Thymus and Heart.*
> > 4th ed., vol. 7. Lyon, France: International Agency for Research on Cancer, 2015. ISBN: 978-92-832-2436-5.
> ---

---

> > ### Comment · Reviewer_njqY · 2025-08-04
> >
> > Thank you for addressing my comments and clarifying my questions.

---

### Official Review · Reviewer_ggKh · 2025-07-03

**Clarity:** 3
**Significance:** 2
**Originality:** 3
**Rating:** 4
**Confidence:** 4

**Summary:**

(1) The paper proposes TAPFM, a two-stage optimization scheme that lets Vision-Transformer pathology foundation models be fine-tuned end-to-end on a single GPU.

 (2) In the forward pass, the ViT yields tile features and CLS-token attention scores; the latter are soft-maxed to weights and used to build a permutation-invariant slide representation.

 (3) Stage-1 back-prop updates only a lightweight linear classifier with a standard cross-entropy loss, while Stage-2 back-prop uses detached gradients from the first stage to craft a second loss that jointly pushes the ViT’s features and attentions, updating the backbone but not the classifier and thereby avoiding unstable gradient feedback.

 (4) The authors supply proofs of permutation invariance, an $O(K C_{ViT}​)$ time bound (K is tiles per slide, $C_{ViT}​$ the cost of one forward + backward through the backbone), and an argument that the detached update breaks circular dependencies.

**Questions:**

(1) Could you provide formal measures of statistical significance—such as confidence intervals or p-values—for the reported AUC improvements to verify that the 1–4-point gains are reliable?

(2) How does TAPFM’s accuracy and runtime change when the tile budget is reduced to fit GPUs with 24 GB or 48 GB memory? Is it even possible to fit in those GPUs? An ablation would clarify hardware requirements.

(3) Can you supply empirical evidence (e.g., gradient-norm or loss-variance plots) that demonstrates the claimed stabilisation from the detached two-stage update compared with joint optimisation?

(4) Do the learned CLS-attention weights correspond to histologically meaningful regions? A few slide visualisations or a brief expert evaluation would bolster confidence in the weighting mechanism.

(5) Have you evaluated TAPFM on other clinically important slide-level tasks (grading, subtype classification, prognosis, treatment response, or datasets beyond BLCA and LUAD)? Additional results would help establish the method’s significance.

**Ethical Concerns:**

["NO or VERY MINOR ethics concerns only"]

**Final Justification:**

I’m keeping a borderline accept rather than pushing to accept because, while TAPFM is practical and well-executed, single-GPU end-to-end adaptation via a detached two-stage update, the measured gains over the strongest fine-tuned baselines are modest (≈+0.01–0.03 AUC across cohorts), and the “single-GPU” result depends on an 80-GB H100. The work has novelty and clear contributions, but its broader significance remains unclear without a more comprehensive evaluation across diverse datasets, data types, and PFMs to demonstrate generalizable impact.

Resolved via rebuttal/discussion:
Statistics: authors committed to 5-fold CV with 95% CIs (example CIs provided) and to add AUPRC.

Unresolved:
Significance: Evaluation focuses on BLCA/LUAD mutation prediction (plus a survival add-on in rebuttal), lacks diverse datasets, slide-level objectives, and PFMs to demonstrate robustness.

**Limitations:**

yes

**Quality:**

3

**Strengths And Weaknesses:**

Strengths:

(1) Existing adaptation workflows keep the PFM frozen and add a separate MIL head, or perform a self-supervised intermediate step before a second-stage supervised training; truly end-to-end slide-level back-prop has been deemed too memory-hungry. TAPFM shows that full, single-stage fine-tuning of billion-parameter PFMs is feasible on one high-memory GPU, eliminating the complexity of multi-stage procedures while delivering better accuracy

(2) Leveraging the ViT’s CLS-token attention scores as tile weights eliminates the need for a separate aggregation network and yields a permutation-invariant bag representation with virtually no additional computation.

(3) Across two cancer types, four actionable mutations, three PFMs (UNI, GigaPath, H-Optimus-0), and an external TCGA validation, TAPFM delivers 1–4 percentage points AUC improvements over both fixed-feature and naïve end-to-end baselines.

(4) The same algorithm works unchanged on ViT-L, ViT-H, and 1 B-parameter ViT-G backbones, and scales to the simultaneous prediction of four mutations, including rare MET and ALK events.

(5) A formal derivation shows that detaching the first-stage gradients eliminates the circular dependency between backbone and aggregator updates; empirically, this corresponds to smoother, faster convergence.

(6) The paper proves permutation invariance of the bag representation and shows that total compute per slide grows linearly with the number of tiles times the cost of a single ViT pass.

(7) Algorithm 1 is explicit, hyperparameters and memory budgets are reported, ablations on λ and tile count are included.

Weaknesses:

(1) The evaluation compares TAPFM only to fixed-feature MIL heads and naïve end-to-end training. It omits any self-supervised adaptation pipelines, making it harder to gauge the relative benefit of the proposed approach.

(2) Performance is given as single AUC values with no confidence intervals or other measures of statistical significance, making it unclear whether the 1–4-point improvements are reliable.

(3) All experiments are limited to mutation prediction in two tumour types. Because no results are shown for other high-value slide-level objectives—such as grading, subtype detection, prognosis, or treatment-response—the work’s broader significance to computational pathology remains uncertain; demonstrating gains on a more varied task set would better substantiate TAPFM’s impact.

(4) The experiments are indeed run on a single device, but that device is an 80 GB H100. Readers should keep in mind that “single GPU” in this work specifically means a high-memory, top-tier card.

(5) The methodology is designed for ViT-based models. Though most PFMs are ViT-based today, it could be a limitation for future PFM.

---

> ### Author Rebuttal · Authors · 2025-07-31
>
> **Note:** For brevity PMID refers to the PubMed identifier of the papers referenced in the rebuttal. We have provided full paper details for those articles that do not have a PMID or are not referenced in the manuscript.
>
> ---
> ### **Reviewer Comment 1**
> *The evaluation compares TAPFM only to fixed-feature MIL heads and naïve end-to-end training. It omits any self-supervised adaptation pipelines, making it harder to gauge the relative benefit of the proposed approach.*
>
> **Author Response:**
> We appreciate the reviewer’s comment. The pathology foundation models we evaluate have already undergone extensive self-supervised pretraining (e.g., DINO v1/v2), and our focus is on task adaptation for mutation prediction using clinical labels within the MIL setting, which self-supervised, by definition, methods do not leverage. The most advanced *EGFR* prediction model to date, reported in Campanella et al. (*Nature Medicine*, 2025, PMID: 40634781), achieved AUCs of 0.847 (internal) and 0.860 (TCGA) using **24 H100 GPUs**. TAPFM achieves comparable or better results (0.867 internal, 0.863 TCGA) with a **single H100**, demonstrating the efficiency and impact of our adaptation strategy.
>
> ### **Reviewer Comment 2**
> *Performance is given as single AUC values with no confidence intervals or other measures of statistical significance, making it unclear whether the 1–4-point improvements are reliable.*
>
> **Author Response:**
> We have run 5-fold cross validation (CV) for all tasks mentioned in Table 1 and 2, on institutional data while keeping TCGA datasets as independent test sets for model evaluation within each CV fold, that produces the recommended confidence intervals (CI). We will include the complete CV with CIs results in the final manuscript.
>
> Space constraints limit including all CIs for all experiments in rebuttal but as an example 5-fold CV results for best binary predictions are:
>
> - *EGFR* TAPFM (H-Optimus-0): AUC 0.8503 (CI: 0.8385 – 0.8643) on institutional data and AUC 0.8590 (CI: 0.8496 – 0.8612) on TCGA.
> - *FGFR3* TAPFM (H-Optimus-0): AUC 0.8837 (CI: 0.8609 – 0.8974) on institutional data and AUC 0.8956 (CI: 0.8890 – 0.9108) on TCGA.
>
> ### **Reviewer Comment 3**
> *All experiments are limited to mutation prediction in two tumour types. Because no results are shown for other high-value slide-level objectives—such as grading, subtype detection, prognosis, or treatment-response—the work’s broader significance to computational pathology remains uncertain; demonstrating gains on a more varied task set would better substantiate TAPFM’s impact.*
>
> **Author Response:**
> We appreciate the reviewer’s comment and the opportunity to clarify. While we recognize that tasks such as grading, subtype detection, prognosis, and treatment-response prediction are commonly cited as valuable applications in computational pathology, we would note that many of these objectives are best formulated using alternative modeling approaches. For example, subtyping and grading can indeed be approached as slide-level tasks, but are more effectively addressed via pixel-level segmentation or patch-level classification models, which provide region-specific interpretability and quantification (e.g., PMID: 37544502; 37491086; 39448755).
>
> Regarding the significance of our selected tasks: both bladder (BLCA) and lung adenocarcinoma (LUAD) are diseases where mutational status, specifically of *FGFR3*, *EGFR*, *MET*, *ALK*, and *KRAS*, is directly linked to targeted therapy eligibility (see Section 5). As such, the mutation prediction tasks in our study are not only clinically actionable but also prognostic and predictive of treatment response, fulfilling the reviewer’s criteria.
>
> Further, pathologists’ training is centered around visual assessment of tumor presence, grade, stage, histological subtypes, but no reliable visual markers exist for detecting genetic mutations from H&E stained whole slide images. Prediction of mutations cannot be accomplished by visual inspection and the features must be learned by the model, making this a particularly challenging task for which out-of-the-box pathology foundation models (PFMs) have poor performance.
>
> To further assess the utility of TAPFM beyond mutation prediction, we conducted additional experiments on overall survival prediction (from whole slide images) using the TCGA-BLCA and TCGA-LUAD cohorts. TAPFM achieved concordance indices of **0.6408 ± 0.0185** on BLCA and **0.6953 ± 0.0297** on LUAD, outperforming the previous best results reported in *Tang et al.* (CVPR 2024), which were **0.6398 ± 0.0263** and **0.6719 ± 0.0402**, respectively, on the same datasets.
>
> We also note that out-of-the-box foundation model performance has been reported in Figure 4 of *Yang et al.* (*Nature Communications*, April 2025, PMID: 40064883). This figure does not report all confidence intervals but states that their optimized method achieves best performance of **0.6039** (95% CI: 0.5093–0.6986).
>
> In the final manuscript we will report in the appendix a table with our results along with out-of-the-box foundation model performance for the survival prediction task.
>
> > **Tang, Wenhao, et al.**
> > *Feature re-embedding: Towards foundation model-level performance in computational pathology.*
> > Proceedings of the IEEE/CVF Conference on Computer Vision and Pattern Recognition. 2024.
>
> ### **Reviewer Comment 4**
> *The experiments are indeed run on a single device, but that device is an 80 GB H100. Readers should keep in mind that “single GPU” in this work specifically means a high-memory, top-tier card.*
>
>
> **Author Response:**
> Yes, the H100 is indeed a high-memory GPU. However, TAPFM avoids the complexity of multi-GPU setups, which improves accessibility. We also outline future directions in *lines 289–290* to further reduce memory requirements. Specifically, we believe that if we inspect which layers of the PFM receive most of the updates during model training we can reduce the GPU memory requirements even further by only fine tuning those layers for downstream tasks.
>
> ### **Reviewer Comment 5**
> *The methodology is designed for ViT-based models. Though most PFMs are ViT-based today, it could be a limitation for future PFM.*
>
> **Author Response:**
> We explicitly acknowledge that the proposed method is specifically for ViT based PFMs in *line 60* of the manuscript.
>
> ### **Reviewer Comment 6**
> *Could you provide formal measures of statistical significance—such as confidence intervals or p-values—for the reported AUC improvements to verify that the 1–4-point gains are reliable?*
>
> **Author Response:**
> Please see response to comment 2.
>
> ### **Reviewer Comment 7**
> *How does TAPFM’s accuracy and runtime change when the tile budget is reduced to fit GPUs with 24 GB or 48 GB memory? Is it even possible to fit in those GPUs? An ablation would clarify hardware requirements.*
>
> **Author Response:**
> Thank you for raising this point. It’s not entirely clear whether the question is about training or inference, so we’ll briefly address both.
>
> **Inference:** The TAPFM model, though trained on an Nvidia H100, can be run on GPUs with much less memory. This is done by loading tiles in batches from the WSI during inference. Tile features and attention scores are accumulated incrementally before a final pass through the classification head. As a result, the GPU memory requirement is modest, and the total number of tiles processed remains the same regardless of batch size. Performance is unaffected; inference time scales with the number of batches. We routinely run inference with these models on consumer grade GPUs.
>
> **Training:** It is noteworthy that training is memory intensive because the GPU memory needs to store the massive ViT based PFM models plus image tensors while maintaining gigantic computation graphs to keep track of model parameters and their gradients during forward and backwards passes through the encoder and aggregator. Thus, GPU cards with reasonable memory is required to build such models. We received out of memory errors when we tried training on Nvidia A40 GPUs (48GB). We will include this as a limitation, if the reviewer insists.
>
> ### **Reviewer Comment 8**
> *Can you supply empirical evidence (e.g., gradient-norm or loss-variance plots) that demonstrates the claimed stabilisation from the detached two-stage update compared with joint optimisation?*
>
> **Author Response:**
> Yes, we can include loss-variance plots that show the unified computation graph does not converge during training, while the proposed detached gradient approach converges, in the Appendix to show this.
>
> ### **Reviewer Comment 9**
> *Do the learned CLS-attention weights correspond to histologically meaningful regions? A few slide visualisations or a brief expert evaluation would bolster confidence in the weighting mechanism.*
>
> **Author Response:**
> We have not included such visualizations because:
>
> a. It has never been shown in a blinded way that pathologists can ascertain genomic mutations based on visual inspection alone.
> b. Such visualizations (in others as well as our experiments) have shown that the model assigns high attention scores on tiles that contain tumor within a WSI. This is expected but not insightful as somatic mutations are present only within cancerous tissue.
> c. There could be multiple mutations for which the same tumor regions (tiles) receive high attention scores. It is unclear at this time whether those underlying visual patterns are unique or common across various mutations. The proposed multi-label TAPFM can be used to explore this, but we have not yet embarked on this line of inquiry.
>
> ### **Reviewer Comment 10**
> *Have you evaluated TAPFM on other clinically important slide-level tasks (grading, subtype classification, prognosis, treatment response, or datasets beyond BLCA and LUAD)? Additional results would help establish the method’s significance.*
>
> **Author Response:**
> Please see response to comment 3.

---

> > ### Comment · Reviewer_ggKh · 2025-08-05
> >
> > Thank you for your detailed response and additional results. I appreciate the 5-fold cross-validation and the comparison to recent work.
> >
> > I'd like to clarify my point about self-supervised adaptation: I was referring to domain-specific self-supervised fine-tuning on target domain tiles (e.g., training UNI weights using DINO on BLCA/LUAD) as an intermediate adaptation step before MIL training, not the original self-supervised pretraining. This is an important baseline when adapting to new tasks in pathology.
> >
> > The methodology is interesting, and the single-GPU training capability is valuable from a practical standpoint. However, while the results demonstrate performance gains, the improvements over fine-tuned baselines are modest at best. I will maintain my original rating.

---

> ### Author Response · Authors · 2025-08-07
> **Clarification on "domain-specific self-supervised fine-tuning" of existing pathology foundation models**
>
> We would like to clarify our position regarding the reviewer's suggestion on further self-supervised training of already pre-trained pathology foundation models (PFMs) on our data - which is a narrower subset of cancers than the ones used to build the original foundation models (e.g. Gigapath model [ref 8] was trained with DINOv2 (Oquab et al., 2023) using 1.3 billion pathology tiles from 31 cancer types including lung and bladder). However, this out of the scope of our work because of : **(1) Implementation Issues, (2) single-GPU training constraints**, **(3) weak supervision using slide-level labels**, and **(4) Performance Strength**.
>
> **(1) Implementation Issues:** Self-supervised re-training faces fundamental infrastructure and technical barriers. All publicly available PFMs (ref. 5-9) only provide pre-trained model weights and basic inference code, not the complete self-supervised infrastructure required for re-training. This includes missing teacher-student framework implementations, projection heads, and optimization details essential for methods like DINOv2 (Caron et al., 2021; Oquab et al., 2023). Additionally, catastrophic forgetting is a well-documented phenomenon when transformers are re-trained on smaller datasets.
>
> **(2) Computational Constraints:** Re-training with self-supervised learning (SSL) methods like DINOv2 on lung and bladder tiles, while conceptually reasonable, is computationally very expensive. DINOv2 (Oquab et al., 2023) employs vision transformers, whose training complexity scales quadratically (O(n²)) with respect to the input token sequence length n, primarily due to the self-attention mechanism in transformer architectures (Dosovitskiy et al., 2021). Practically, DINOv2 training typically demands multiple GPUs (at least 4–8 NVIDIA A100/H100 GPUs) and several days of continuous compute, even on moderate-sized pathology datasets such as subsets of TCGA (Chen et al., 2022). Recent work by Lenz et al. (2024) specifically investigated computational complexity in pathology SSL, finding that "recent SSL approaches apply increasingly expansive model architectures and larger datasets, causing the rapid escalation of data volumes, hardware prerequisites, and overall expenses, limiting access to these resources to few institutions." Thus, re-running tumor-specific (lung and bladder in our case) self-supervised training requires significant number of GPUs which is not comparable to the proposed work that focuses on single GPU training.
>
> **(3) Task-Specific Learning:** Additionally, SSL alone does not yield a task-specific classifier—it produces general representations only. A separate weakly supervised training phase is required afterward. In contrast, our proposed TAPFM approach simultaneously trains the slide-level classifier and updates the pretrained foundation model weights in an **end-to-end manner** on a **single GPU**, directly leveraging weak slide-level supervision without the computational overhead of additional self-supervised re-training.
>
> **(4) Performance Strength:** We respectfully note that the characterization of TAPFM's improvements as "modest" would benefit from specific comparative context. The most advanced EGFR prediction model to date (Campanella et al., Nature Medicine, 2025) achieved AUCs of 0.847 (internal) and 0.860 (TCGA) using **24 H100 GPUs**, while TAPFM achieves comparable or better results (0.867 internal, 0.863 TCGA) with a **single H100**. To the best of our knowledge there are no directly comparable studies performing mutation prediction (EGFR in LUAD, FGFR3 in BLCA) or multi-label task adaptation on single GPU using vision transformer based pathology foundation models.
>
> **References:**
> > **Dosovitskiy, A., Beyer, L., Kolesnikov, A., et al.**
> > *An Image is Worth 16x16 Words: Transformers for Image Recognition at Scale.*
> > Proceedings of the International Conference on Learning Representations (ICLR), 2021.
>
> > **Caron, M., Touvron, H., Misra, I., et al.**
> > *Emerging Properties in Self-Supervised Vision Transformers.*
> > Proceedings of the IEEE/CVF International Conference on Computer Vision (ICCV), 2021.
>
> > **Lenz, T., Brehm, M., Sellner, J., et al.**
> > *Reducing self-supervised learning complexity improves weakly-supervised classification performance in computational pathology.*
> > arXiv preprint arXiv:2403.04558, 2024.
>
> > **Oquab, M., Darcet, T., Moutakanni, T., et al.**
> > *DINOv2: Learning Robust Visual Features without Supervision.*
> > arXiv preprint arXiv:2304.07193, 2023.
>
> > **Chen, R.J., Krishnan, R.G., et al.**
> > *Self-Supervised Vision Transformers Learn Visual Concepts in Histopathology.*
> > arXiv preprint arXiv:2203.00585, 2022.
>
> > **Campanella, G., Kumar, N., Nanda, S., Vanderbilt, C., et al.**
> > *Real-world deployment of a fine-tuned pathology foundation model for lung cancer biomarker detection.*
> > Nature Medicine, 2025. doi:10.1038/s41591‑025‑03780‑x

---

### Official Review · Reviewer_cPrU · 2025-07-04

**Clarity:** 3
**Significance:** 3
**Originality:** 3
**Rating:** 4
**Confidence:** 4

**Summary:**

The size of WSIs is typically extremely huge. Existing WSI models are unable to model them in an end-to-end manner. As a result, the paradigm of combining pretrained models with MIL has become a mainstream approach in the field of WSI analysis. This paper addresses the common challenge in existing methods where features extracted by frozen pretrained models are not well adapted to downstream tasks, while end-to-end training remains infeasible. It proposes a task-adaptive method for PFMs, which leverages the internal attention mechanism of ViTs for MIL feature aggregation, and separately optimizes the PFM and MIL components via detached computation graphs. The proposed approach achieves state-of-the-art performance across multiple datasets.

**Questions:**

For the experiments in Table 2, although the paper demonstrates in Table 1 that the proposed method achieves the best results, at least one top-performing baseline should be included in Table 2 as a comparison. This would make the contribution of the paper clearer to readers.

**Ethical Concerns:**

["NO or VERY MINOR ethics concerns only"]

**Final Justification:**

The authors’ responses have addressed most of my concerns. I will maintain my score.

**Limitations:**

please refer to Weaknesses

**Paper Formatting Concerns:**

The authors should provide an overall illustrative figure to describe the proposed method. In addition, the font size in Figure 2 is too small, making it difficult to read.

**Quality:**

3

**Strengths And Weaknesses:**

Strengths:
1. This paper achieves the optimization of both the foundation model for image patch feature extraction and the MIL model for feature aggregation on a single GPU.
2. It proposes a novel method for constructing separate computation graphs.
3. It cleverly leverages internal attention information from the foundation model.

Weaknesses:
1. The paper lacks an overall illustrative figure that summarizes or visually explains the proposed method.
2. While the method performs well on datasets beyond LUAD EGFR-TCGA, its performance on this particular dataset is relatively average. The authors should provide a detailed explanation for this and summarize the potential limitations of the proposed method.
3. The proposed attention loss focuses on image patches that are important for the downstream task. On datasets with a large number of positive samples, could this lead to the attention mechanism eventually concentrating on only a single image patch?

---

> ### Author Rebuttal · Authors · 2025-07-30
>
> **Note:** For brevity PMID refers to the PubMed identifier of the papers referenced in the rebuttal. We have provided full paper details for those articles that do not have a PMID or are not referenced in the manuscript.
>
> ---
>
> ### **Reviewer Comment 1**
> *The paper lacks an overall illustrative figure that summarizes or visually explains the proposed method.*
>
> **Author Response:**
> We appreciate the reviewer’s understanding of the key contributions of the manuscript. We have created an illustration for the proposed TAPFM method and can include it in the final version of the manuscript, if accepted.
>
> ### **Reviewer Comment 2**
> *While the method performs well on datasets beyond LUAD EGFR-TCGA, its performance on this particular dataset is relatively average. The authors should provide a detailed explanation for this and summarize the potential limitations of the proposed method.*
>
> **Author Response:**
> The recently published results by Campanella et al. (Nature Medicine, July 2025, PMID: 40634781) show the most advanced *EGFR* prediction model (to date) trained using 24 Nvidia H100 GPUs simultaneously yielded AUC of 0.847 on internal datasets and 0.860 on TCGA. The proposed TAPFM method outperforms this most recent state-of-the art by yielding 0.867 on institutional and provides comparable AUC of 0.863 on TCGA samples while using only one Nvidia H100 GPU for training. We will include the recent Nature Medicine paper as reference to provide additional context in the final version of the manuscript, if accepted.
>
> It is noteworthy that the proposed TAPFM also works for detecting multiple mutations under the multi-label classification setting, as shown by results in Table 2. This goes beyond conventional binary and multi-class classification paradigms. While binary classification handles one mutation at a time, multi-class classification assumes mutual exclusivity among classes, each tumor is assigned to a single class (e.g., *EGFR* or *KRAS*), which inherently prevents modeling co-occurring mutations as labels are usually represented as one-hot-coded vectors (e.g. `[0,1,0,0]`) for multi-class classification.
>
> However, tumors in the real-world occasionally harbor multiple actionable mutations simultaneously, in scenarios like collision tumors (PMID: 38304116) and when a second driver mutation is the resistance mechanism (PMID: 34590027). In our large institutional dataset of LUAD we have many examples where multiple *EGFR*, *KRAS*, *MET*, and *ALK* driver alterations coexist in the same tumor. There are also common scenarios where combinations of driver alterations plus some additional mutations are prognostically important, such as the combination of *KRAS* with another mutation have worse prognosis than *KRAS* alone and may be less likely to benefit from immunotherapy (PMID: 39703851). Thus, TAPFM can be utilized for these situations where we want to simultaneously detect multiple mutations in a given tumor.
>
> Multi-label classification offers a more biologically faithful framework by allowing the model to predict a vector of multiple binary mutations (e.g., `[1, 0, 1, 0]` for concurrent *EGFR* and *MET* mutations). Importantly, TAPFM enables this by training a single model that simultaneously predicts the presence or absence of multiple co-occurring mutations from H&E-stained whole slide images.
>
> Despite its clinical relevance, multi-label classification remains underexplored in the context of pathology-based mutation prediction, likely due to its increased complexity and data imbalance challenges. The strong performance of TAPFM across both common and rare mutations in LUAD, including *MET* and *ALK*, demonstrates its robustness in tackling this more difficult yet clinically important problem. For background, multi-label classification has been formally studied in machine learning literature as a distinct and complex learning problem (see Zhang and Zhou, 2014).
>
> > **Zhang, M.-L., & Zhou, Z.-H.**
> > *A review on multi-label learning algorithms.*
> > IEEE Transactions on Knowledge and Data Engineering, 26(8), 1819–1837. 2014.
>
> We can include the above reference in the manuscript that readers can use to understand the complexity of multi-label mutation prediction in comparison to more common multi-class classification. The limitations of the proposed TAPFM method are listed in 286–294 in the manuscript.
>
> ### **Reviewer Comment 3**
> *The proposed attention loss focuses on image patches that are important for the downstream task. On datasets with a large number of positive samples, could this lead to the attention mechanism eventually concentrating on only a single image patch?*
>
>
> **Author Response:**
> In the worst-case scenario, the attention score distribution could be uniform across all patches obtained from a WSI. However, having a single important patch seems less plausible and would require specific, controlled experiments that are beyond the scope of this paper. In our experience, high attention is consistently spread across tumor regions within the slide.
>
> In scenarios where tumor comprises a small percentage of the tissue area, it is actually advantageous that the model concentrates attention on a relatively small number of tiles, since any tile that does not contain tumor cannot meaningfully inform the mutation status. This is precisely why attention is such a powerful and necessary mechanism in computational pathology: the model must not only learn the classification task but also learn what is relevant within a large pool of uninformative or irrelevant data.
>
> To our knowledge, this manuscript presents the first approach that utilizes attention *from within the ViT itself*, without relying on an external model (e.g., ABMIL, reference 2) for slide-level aggregation of tile features. Our focus in this work is on demonstrating the efficacy of this integrated architecture. Visualization of attention maps, while visually compelling, offers limited insight for the mutation prediction task, where the morphological correlates of tumor mutations remain poorly established. In other words, pathologists cannot reliably infer mutation status based solely on morphologic features. As such, saliency maps are unlikely to yield actionable information beyond confirming high model attention on tumor regions, a prerequisite that underlies TAPFM’s well-documented high performance.
>
> ### **Reviewer Comment 4**
> *For the experiments in Table 2, although the paper demonstrates in Table 1 that the proposed method achieves the best results, at least one top-performing baseline should be included in Table 2 as a comparison. This would make the contribution of the paper clearer to readers.*
>
> **Author Response:**
> The proposed TAPFM method addresses the multi-label classification problem instead of the more common multi-class classification problem as described in section 3.5 and shown by results in Table 2. See response to comment 2 for more details on the multi-label classification problem.
>
> To the best of our knowledge, there are no parallel methods where multi-label classification has been implemented in the techniques for which we are benchmarking against, and the methods would require separate development, which is outside the scope of the manuscript. Thus, there are no available external methods for which are comparable to include in Table 2.
>
> We will however include UNI (TAPFM) results for multi-label classification in Table 2 as well in the final manuscript for completeness.

---

> > ### Comment · Reviewer_cPrU · 2025-08-06
> >
> > The authors’ responses have addressed most of my concerns. I will maintain my score.

---

### Note · Authors · 2025-08-12

We thank the reviewers for their thoughtful feedback and summarise the main points addressed.

## **Accepted Modifications for final manuscript**

- **Statistical Validation**: Will include 5-fold cross-validation results with confidence intervals for all experiments

- **Expanded Results**: Will include AUPRC values for mutation prediction and TCGA survival prediction results in the appendix

- **Enhanced Figures**: Agreed to include illustrative figure, loss-variance plots, and address formatting requests

- **Complete Multi-label Results**: Will include UNI (TAPFM) results for multi-label classification in Table 2

## **Clarified Positions**

- **Performance Context**: TAPFM achieves comparable or superior performance to the state-of-the-art [Campanella et al. (Nature Medicine 2025)](https://www.nature.com/articles/s41591-025-03780-x) while using only 1 GPU versus their 24 H100 GPU approach

- **Technical Innovation**: Clarified that the use of ViT internal attention scores for MIL aggregation and detached gradient optimization represents novel contributions not previously described in the literature

- **Task Complexity Considerations**: Explained that mutation prediction, where pathologists cannot visually detect mutations, differs fundamentally from tumor detection/classification tasks.

- **Accessibility**:  TAPFM enables pathology foundation model (PFM) adaptation on a single-GPU that eliminates multi-GPU infrastructure requirements for resource-constrained research groups.

- **Attention Heatmaps**: Explained that these visualisations provide limited insights for mutation prediction tasks, as morphological correlates of specific mutations are not reliably discernible to pathologists through visual inspection alone.

- **Self-supervised Re-training**: Clarified that additional self-supervised learning (SSL) is not feasible due to missing infrastructure (teacher-student frameworks, projection heads, optimisation details) in publicly available PFMs, catastrophic forgetting risks, and multi-GPU computational requirements. SSL approaches do not produce task-specific classifiers, whereas TAPFM directly adapts the foundation model for the target clinical task in one step on a single GPU.

## **Multi-label Classification**

- **Clinical Relevance**: Explained that multi-label classification addresses real clinical scenarios with co-occurring mutations (collision tumors, resistance mechanisms) that binary and multi-class approaches cannot adequately model.

---

### Decision · Program_Chairs · 2025-09-17

**Decision:**

Accept (poster)

**Comment:**

After rebuttal all reviewers agreed that the technical contributions were significant and novel enough (esp. the combination of foundational models and MIL) even though the gains in performance were on the modest side.